CellPress

## Perspective

# Monogenic disorders of immunity: Common variants are not so rare

Vivien Béziat[1,2,3,*] and Jean-Laurent Casanova[1,2,3,4,5,*]

[1]Laboratory of Human Genetics of Infectious Diseases, Necker Branch, INSERM, Necker Hospital for Sick Children, Paris, France
[2]Imagine Institute, Paris-Cité University, Paris, France
[3]St. Giles Laboratory of Human Genetics of Infectious Diseases, Rockefeller Branch, The Rockefeller University, New York, NY, USA
[4]Pediatric Hematology-Immunology and Rheumatology Unit, Necker Hospital for Sick Children, AP-HP, Paris, France
[5]Howard Hughes Medical Institute, The Rockefeller University, New York, NY, USA
*Correspondence: vivien.beziat@inserm.fr (V.B.), casanova@rockefeller.edu (J.-L.C.)

## SUMMARY

According to the current paradigm, human monogenic disorders underlying immunological phenotypes are due to rare (frequency <1%) as opposed to common (>1%) alleles. However, as reviewed here, an increasing number of studies have reported monogenic disorders of immunity, recessive or dominant, involving alleles that are currently common in specific small or large populations. Examples range from *IFNAR1* and *IFNAR2* null alleles in the Arctic and Pacific to *PTCRA* hypomorphic alleles in South Asia. This situation may be explained by a history of (1) population bottlenecks followed by expansion; (2) genetic drift before the advent of an environmental trigger; (3) slow purging, especially for recessive, mild, or incompletely penetrant conditions; and/or (4) balancing selection with a heterozygous advantage. In patients with suspected monogenic immunological conditions, a role for alleles common in the corresponding population should not be excluded. At odds with the prevailing view, common alleles may underlie monogenic disorders of immunity and should therefore be considered.

## INTRODUCTION

Human inborn errors of immunity (IEIs) were originally described, during the period from 1946 to 1952,[1–3] as Mendelian traits—monogenic disorders with complete penetrance. If not all carriers of an at-risk genotype display the corresponding phenotype, then penetrance is incomplete, and monogenic inheritance can be referred to as "non-Mendelian." Since the late 1990s, the description of incomplete penetrance for disease-causing immunological phenotypes, and even more frequently for the clinical phenotypes themselves, has progressively blurred the initially described Mendelian nature of these conditions. A number of "non-Mendelian monogenic disorders" have been described for infection, autoinflammation, autoimmunity, allergy, and cancers.[4] When at least two kindreds are available, the relative risk (RR), or odds ratio (OR) of developing a phenotype of interest can be estimated and should be higher than an arbitrary but conservative threshold in carriers of the at-risk genotype. The concepts of RR and OR, as well as the relationship between RR and penetrance, are explained in Box 1. Gaining an understanding of the mechanisms underlying incomplete penetrance of monogenic disorders is a major endeavor in the field of IEI and human genetics at large. A handful of studies have yielded promising results in this respect[5–7] and are detailed in Box 2.

Here, we postulate that a disease can be considered monogenic even with low penetrance, provided that it is driven by a monogenic genotype, as supported by genetic data (i.e., associ-ation of genotype and phenotype in a multiplex kindred or, better, in multiple kindreds) or experimental (i.e., molecular and cellular mechanism connecting genotype and phenotype, which is required for single-patient studies and preferable for single-kindred studies).[19,20] This definition does not exclude the possi-bility that an allele conferring high risk with low penetrance may require as yet unknown alleles at a modifier locus to underlie the phenotype, as explained in Box 2. Admittedly, the distinction be-tween monogenic and digenic can become arbitrary; the respec-tive contributions of the alleles at the two loci (e.g., their fre-quency, functional impact, or both) may tip the balance in favor of either term. Moreover, there is no universally accepted "monogenic threshold," but an OR/RR $\geq$ 5 seems to be a reasonably conservative threshold in this context, both biologi-cally and clinically, and is used in this review.

Most monogenic disorders of immunity were discovered via patient- and family-based studies or studies of rare conditions in the field of IEI. However, a subset of monogenic immunolog-ical conditions was discovered in large population-based studies.[21–25] In these studies, focusing on common conditions, the involvement of a common allele was expected. By contrast, regardless of their penetrance, patient-based IEIs have been widely considered, since the discovery of the first genotypes in 1985,[26,27] as being due to rare alleles, with a minor-allele frequency (MAF) of <0.01 across the populations studied. This notion is consistent with the rarity of the original IEI-defining traits themselves. Consequently, most exome or

---

### Box 1. Effect size, RR, OR, and penetrance

The effect size quantifies how much a genetic variation influences a particular characteristic, such as disease risk. RR and OR are classical measures of effect size of a variant for a binary trait (e.g., affected/unaffected). The RR between groups is calculated as the ratio of the penetrance of the phenotype of interest between the two groups. The penetrance in a group (e.g., carriers or non-carriers of an at-risk genotype) is the observed frequency of the phenotype in the group and is denoted here as $f$. If group A (carriers) has a penetrance of $f_a$ and group B (non-carriers) a penetrance of $f_b$, then the RR of the phenotype in group A relative to group B is $f_a/f_b$. If, for example, $f_a = 0.2$ and $f_b = 0.01$, then RR = 20. In some circumstances, there is a linear relationship between the RR and $f_a$.[8] By contrast, the OR between groups is calculated as the ratio of the odds of the two groups. The odds of group A are $f_a/(1-f_a)$. The odds of group B are $f_b/(1 - f_b)$. Thus, the OR of group A relative to group B is $(f_a/(1 - f_a))/(f_b/(1 - f_b))$. In our example, the OR = $(0.2/0.8)/(0.01/0.99) = 24.75$. The RR should be preferred over the OR where possible, but, unlike the OR, it is often impossible to calculate in case-control studies. For rare events (e.g., diseases), the OR usually provides a good approximation of the RR. An OR or RR close to 1 indicates a lack of association between the genotype and phenotype considered. A low or high OR or RR, deviating significantly from 1, is suggestive of a negative or positive association, respectively. Nevertheless, neither implies causality or provides any information about the underlying mechanism. Causality can be inferred from existing biological or medical knowledge or from additional, mechanistic experiments at the molecular, cellular, or whole-organism level.

---

### Box 2. Possible causes of incomplete penetrance for monogenic disorders

Documented and suggested mechanisms of incomplete penetrance include (1) environmental factors, as a lack of or insufficient exposure to environmental triggers, including pathogens, allergens, and carcinogens, can obviously account for a lack of phenotype in at-risk individuals (e.g., in IL-12Rβ1-deficient individuals, BCG disease cannot occur in the absence of BCG vaccination[9]); (2) broad, pre-existing adaptive immunity to the same or a related pathogen, as the recognition of the invading microbe by T or B cells can mitigate a genetic deficiency of innate or intrinsic immunity (e.g., BCG disease protects IL-12Rβ1-deficient individuals against environmental mycobacterial disease[9]); (3) narrow, pre-existing humoral responses to a specific microbial virulence factor can even mask the innate genetic disorder (e.g., antibodies against lipoteichoic acid [LTA] prevent staphylococcal disease in TIR Domain Containing Adaptor Protein [TIRAP]-deficient individuals, whose defect renders them susceptible exclusively to LTA-expressing staphylococci[6,7]); (4) non-random monoallelic expression, which can lead to expression of the wild-type and mutant alleles in healthy and sick heterozygotes, respectively (e.g., selective expression of the mutant *Phospholipase C Gamma 2* [*PLCG2*] allele in the B cells of affected heterozygous carriers, leading to antibody deficiency[10]); (5) somatic mosaicism, as the reversion of the germline defect in relevant cell lineages can improve the condition (e.g., reversion to wild-type of one mutant allele in the B cells of a patient with ADA deficiency progressively improves the clinical phenotype[11]); (6) modifier genes, as germline epistasis can govern the penetrance of single-gene lesions (e.g., a common allele of *Bone Morphogenetic Protein 2* [*BMP2*] greatly increases the risk of developing craniosynostosis in patients with deleterious *SMAD Family Member 6* [*SMAD6*] alleles[12–15]); and (7) and regulatory variants in *cis* of a heterozygous variant that influence the expression level of the mutant or wild-type allele.[16–18]

---

genome pipelines in both research and diagnostic laboratories performing analyses on individual patients and families currently filter out common alleles, defined as those with a MAF >0.01. Here, we adopted a MAF of 0.01 as the cutoff between rare and common alleles, even though such a threshold is arbitrary, because this is the most widely used threshold and, as such, the most appropriate for our purpose of revisiting its usefulness. These filters consider the global MAF, or the highest MAF, or the MAF in the corresponding population—typically one of the seven major ancestries, more rarely a smaller population.

Of course, there is also no "size threshold" for defining a population, and one could even provocatively assert that any disease-causing allele is common in any affected kindred, treated as an ultra-small population. More reasonably, a hamlet or a village could justifiably be considered to contain a population, particularly if geographically isolated—on a small island or in a mountain valley, for example. We review here the known common alleles underlying monogenic immunological conditions, including both IEIs identified in patient-based studies and conditions identified in studies of populations (Figure 1). The high frequency of theses alleles may be theoretically explained by a history of (1) population bottlenecks followed by expansion; (2) genetic drift before the advent of an environmental trigger; (3) slow purging, especially for recessive, mild, or incompletely penetrant conditions; and/or (4) balancing selection with a heterozygous advantage. In most cases, their high frequency remains unexplained. We divide these variants into two groups, based on high and low penetrance, in which they are classified chronologically.

## PATHOGENIC ALLELES WITH HIGH PENETRANCE

### Common variants of *C2* can underlie infection or autoimmunity

Patient-based studies led to the discovery of common genetic defects of complement.[28] There are three complement activation pathways: the classical, alternative, and lectin pathways.[29] The classical pathway is activated by the Fc fragment of an immunoglobulin (IgG1, IgG2, IgG3, or IgM) linked to an antigen. The lectin pathway, homologous to the classical pathway, is activated by MBL (mannose-binding lectin) or ficolins, which recognize mannose residues on the surface of pathogens. The alternative pathway is directly activated by binding of the C3 protein to pathogens. Each pathway has its own cascade of consequences, but all three converge to activate the terminal pathway, leading to formation of the membrane attack complex (MAC). MAC assembly results in the formation of pores in the lipid membranes of pathogens, particularly Gram-negative bacteria, leading to their lysis. Genetic deficiencies of MAC components underlie invasive disease due to *Neisseria*.[28,29] Complete deficiencies of alternative pathway proteins confer a predisposition to invasive bacterial infections.[28,29] Deficiencies of the lectin pathway are not associated with a well-defined clinical phenotype.[30] Deficiencies of the classical pathway underlie invasive bacterial infections and systemic lupus erythematosus.[31,32]

Deficiency of C2, a crucial molecule in the lectin and classical complement pathways, was first described in 1966.[33] It was later

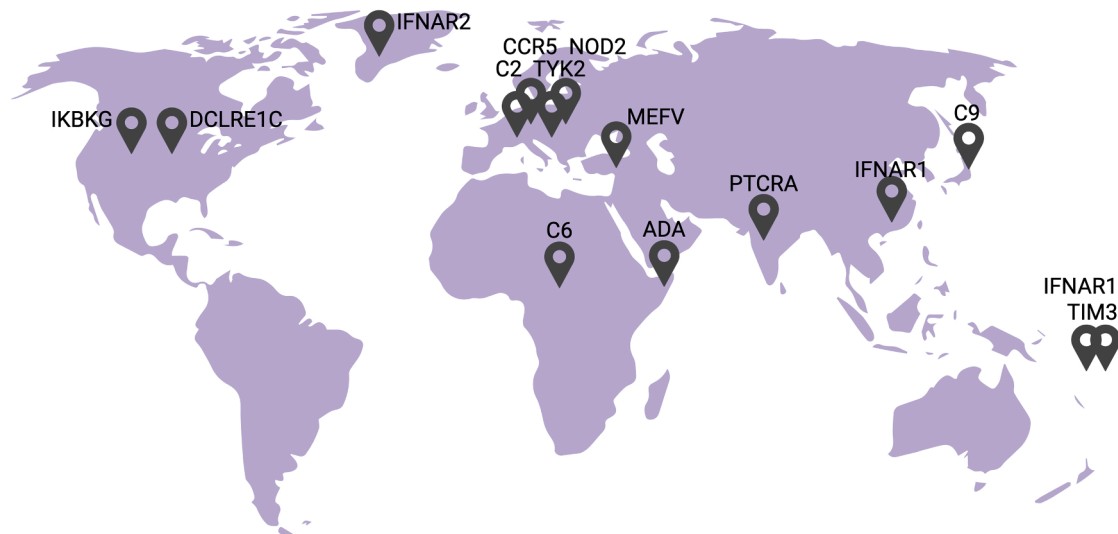

**Figure 1. Worldwide distribution of frequent variants underlying IEIs**
Complete C2 deficiency underlies invasive bacterial infection and systemic lupus erythematosus and is frequent in Europeans. Complete C6 and C9 deficiencies underlie invasive *Neisseria* disease and are frequent in sub-Saharan Africans and Japanese, respectively. MEFV GoF variants underlie FMF and are particularly prevalent in Arabs, Armenians, Jews, and Turks. Complete NOD2 deficiency underlies Crohn's disease with low penetrance and is frequent in Europeans. Complete deficiencies of ADA, DCLRE1C, and IKBKG underlying SCID are frequent in Somalians, Native Americans (Apache and Navajo) and Northern Cree Canadians, respectively. CCR5 deficiency is frequent in Europeans, where it protects against HIV-1 infection but confers a predisposition to WNV disease. Partial TYK2 deficiency underlies TB and is frequent in Europeans. Complete TIM3 deficiency underlies SPTCL and HLH with low penetrance and is frequent in East Asians and Polynesians. Complete deficiencies of IFNAR1 and IFNAR2 underlie severe viral infections and are frequent in Western Polynesians and Arctic populations, respectively. Partial IFNAR1 deficiency underlies viral infections and is frequent in South Han Chinese. Partial PTCRA deficiency underlies autoimmune phenotypes and is frequent in South Asia and the Middle East.

recognized as the most common complement protein deficiency in the European population.[34] In 90% of cases, it is caused by a deletion of 28 bp (rs9332736) in *C2*, leading to skipping of exon 6 and a complete absence of C2 protein synthesis.[35] The allele frequency for this deletion is ∼0.007 in Europeans and up to 0.01 in Ashkenazi Jews, according to gnomAD v.4.1[36] (Table 1). The clinical penetrance of C2 deficiency is incomplete and may be overestimated due to observation bias.[37] In a Swedish series of 40 C2-deficient patients, 34 patients were homozygous for the 28 bp deletion (rs9332736) in *C2*, and three others were compound heterozygous for this and another deleterious variant. About 25% of the patients had systemic lupus erythematosus or a related autoimmune disease, and about 60% of them had a history of invasive infection. For the individuals with invasive infection, 64.3% of the cases of meningitis and 52.2% of the cases of septicemia were due to *Streptococcus pneumoniae*.[38] Three patients in this series (7.5%) had a history of meningococcal disease—meningitis in two cases and sepsis in the third. Over a mean observation time of 39 years, six of the 40 patients (15%) developed only minor infections and had no autoimmune phenotype at their last follow-up visit, suggesting a clinical penetrance of at least 85% in a lifetime (Table 2).

### Common variants of *C6* or *C9* can underlie infection

MAC proteins include the C5, C6, C7, C8α, C8β, C8γ, and C9 proteins.[81] Inherited deficiencies of all MAC proteins except C8γ have been reported in humans. Hundreds of patients with MAC deficiency have been reported since the description of the first case in 1976.[82–84] Individuals with deficiencies of the C5-C8 MAC proteins present a narrow spectrum of susceptibility to infection, limited to bacteria of the genus *Neisseria* (*Neisseria meningitidis* and *Neisseria gonorrhoeae*)[41] and, more rarely, *Haemophilus parainfluenzae*.[85,86] These genotypes result in a complete inability to form the MAC and an increase in the risk of infection by a factor of 7,000–10,000 with at least one episode of meningococcal disease in 50%–60% of at-risk genotype carriers (Table 2).[41,69] C6 deficiency was first described in 1974 in a single kindred,[87] and a high prevalence of C6 deficiency in African-Americans was reported in 1984.[41] Three variants of *C6* account for most of the MAC protein deficiencies reported in Africans: p.Gln274Argfs*46 (rs557023458), p.Gln380Serfs*7 (rs375762365), and p.Asp627Thrfs*4 (rs61469168).[42,88,89] The allele frequencies of these variants in Africans and African Americans in gnomAD v.4.1 are 0.0042, 0.0068, and 0.0108, respectively (Table 1). Based on the cumulative frequencies of the three major alleles, 1 in 2,500 Africans has a complete deficiency of C6. As a result, the African population has a particularly strong predisposition to meningococcal meningitis. Indeed, Africa, with its sadly famous "meningitis belt," is well known to be the region of the world with the highest prevalence of meningitis and an incidence of up to 1 in 1,000.[90,91]

C9 deficiency was first described in 1979 in a single Japanese kindred.[92] C9 deficiency confers a predisposition to invasive *Neisseria* disease but to a lesser extent than other terminal pathway deficiencies affecting the C5–C8 proteins.[43,44] Indeed, during formation of the MAC, C9 allows enlargement of the pore but is not

**Table 1. Common alleles underlying monogenic predispositions to infection, autoimmunity, or autoinflammation**

| Gene (product)[a] | SNP | Chr. | Position (GRCh38) | Reference | Alt. | Consequence (Mane selected transcript) | Heritability | Populations with highest MAF | Highest MAF | Clinical phenotype | Reference |
|---|---|---|---|---|---|---|---|---|---|---|---|
| *ADA* | rs1057520217 | 20 | 44651601 | G | A | p. Gln3* | recessive | Somalians | 0.024 | SCID | Sanchez et al.,[39] Adams et al.[40] |
| *C2* | rs9332736 | 6 | 31934288 | ATGGTG GACAGG GTCAGG AATCAG GAGTC | A | skipping of exon 6 | recessive | Ashkenazi Jewish (gnomAD v.4.1) | 0.01135 | *Neisseria* meningitis (high penetrance) | Johnson et al.[35] |
| *C6* | rs61469168 | 5 | 41158762 | TC | T | p.Asp627Thrfs*4 | recessive | African (gnomAD v.4.1) | 0.01080 | *Neisseria* meningitis (high penetrance) | Ross et al.,[41] Nishizaka et al.[42] |
| *C9* | rs121909592 | 5 | 39341276 | G | A | p.Arg116* | recessive | Japanese | 0.0298 | *Neisseria* meningitis (low penetrance) | Nagata et al.,[43] Nishizaki et al.,[44] Kira et al.,[45] Higasa et al.[46] |
| *CCR5* | rs333 | 3 | 46373452 | TACAGT CAGTAT CAATTC TGGAAG AATTTC CAG | T | p.Ser185Ilefs*32 | recessive Mendelian | European | 0.1315 (gnomAD v.4.1) | WNV (+resistance to HIV-1 infection) | Glass et al.,[24] Lim et al.,[25] Dean et al.,[47] Liu et al.,[48] Samson et al.[49] |
| *DCLRE1C* (ARTEMIS) | rs121908157 | 10 | 14934461 | G | T | p.Tyr199* | recessive | Navajo and Apache Native Americans | 0.021 | SCID | Murphy et al.,[50] Li et al.[51] |
| *HAVCR2* (TIM-3) | rs184868814 | 5 | 157106776 | T | C | p.Tyr82Cys | recessive | East Asians (gnomAD v.4.1) | 0.01656 | HLH and SPTCL | Gayden et al.[52] |
| *IFNAR1* | rs72552343 | 21 | 33349398 | TTCC | T | p.Pro335del | recessive (possibly dominant) | Southern Han Chinese | 0.011–0.023 | viral infections | Al Qureshah et al.,[53] Zhang et al.[54] |
| *IFNAR1* | rs201609461 | 21 | 33352770 | G | T | p.Glu386* | recessive | Western Polynesians | 0.01250 | viral infections | Bastard et al.[55] |
| *IFNAR2* | rs1987287426 | 21 | 33245010 | T | C | p.Ser53Pro | recessive | Inuits | 0.034 | viral infections | Duncan et al.[56] |
| *IKBKB* | rs886041036 | 8 | 42318603 | G | GG | p.Gln432Profs*62 | recessive | Northern Cree | 0.076 | SCID | Pannicke et al.,[57] Rubi et al.[58] |
| *MEFV* | rs61752717 | 16 | 3243407 | T | C | p.Met694Val | recessive (possibly dominant) | Turkish[b] North African Jews | 0.017 0,08 | FMF | French FMF Consortium,[59] The International FMF Consortium et al.,[60] Shohat et al.,[61] Stoffman et al.,[62] Honda et al.[63] |

*(Continued on next page)*

**Table 1.  *Continued***

| Gene (product)[a] | SNP | Chr. | Position (GRCh38) | Reference | Alt. | Consequence (Mane selected transcript) | Heritability | Populations with highest MAF | Highest MAF | Clinical phenotype | Reference |
|---|---|---|---|---|---|---|---|---|---|---|---|
| *MEFV* | rs28940579 | 16 | 3243310 | A | G | p.Val726Ala | recessive (possibly dominant) | Ashkenazi Jews (gnomAD v.4.1) | 0.03914 | FMF | Johnson et al.,[35] French FMF Consortium,[59] The International FMF Consortium et al.,[60] Stoffman et al.,[62] Honda et al.[63] |
| *NOD2* | rs2066844 | 16 | 50712015 | C | T | p.Arg675Trp | recessive | non-Finnish Europeans (gnomAD v.4.1) | 0.04766 | Crohn's disease | Ogura et al.,[21] Hugot et al.,[22] Hampe et al.,[23] Ahmad et al.,[64] Bonen et al.[65] |
| *NOD2* | rs2066845 | 16 | 50722629 | G | C | p.Gly881Arg | recessive | Middle East (gnomAD v.4.1) | 0.03728 | Crohn's disease | Ogura et al.,[21] Hugot et al.,[22] Hampe et al.,[23] Ahmad et al.,[64] Bonen et al.[65] |
| *NOD2* | rs2066847 | 16 | 50729867 | G | GC | p.Leu980Profs*2 | recessive | non-Finnish Europeans (gnomAD v.4.1) | 0.02163 | Crohn's disease | Ogura et al.,[21] Hugot et al.,[22] Hampe et al.,[23] Ahmad et al.,[64] Bonen et al.[65] |
| *PTCRA* (Pre-TCRα) | rs200942121 | 6 | 42923120 | A | C | p.Asp51Ala | recessive | South Asians (gnomAD v.4.1) | 0.01675 | various autoimmune diseases | Materna et al.[66] |
| *TYK2* | rs34536443 | 19 | 10352442 | G | C | p.Pro1104Ala | recessive | non-Finnish Europeans (gnomAD v.4.1) | 0.04450 | TB | Boisson-Dupuis et al.,[67] Kerner et al.[68] |

[a]HLA alleles are not included.

[b]Turkiye National Genome and Bioinformatics Project. Turkish Genome Project Data Sharing Portal v.1.0 (tgd.tuseb.gov.tr/en). Accessed May 23, 2024.

CelPress

**Table 2. Penetrance and RR of IEIs associated with common variants**

| Gene | Type of deficiency | Transmission model | Penetrance | RR or OR | Reference |
|---|---|---|---|---|---|
| ADA | complete | recessive | 100% | N.D. | Sanchez et al.,[39] Adams et al.[40] |
| C2 | complete | recessive | >85% (lifetime) | N.D. | Jönsson et al.[38] |
| C6 | complete | recessive | 50%–60% | RR = 7,000 to 10,000 | Ross et al.,[41] Figueroa et al.[69] |
| C9 | complete | recessive | 5%–10% (lifetime, Fukuoka city) | RR ∼ 1,000 | Nagata et al.[43] |
| CCR5 | complete | recessive | N.D. | OR = 5 | Glass et al.,[24] Lim et al.,[25] Ellwanger et al.[70] |
| DCLRE1C (ARTEMIS) | complete | recessive | 100% | N.D. | Li et al.[71] |
| HAVCR2 (TIM-3) | complete | recessive | max 0.3%/year[a] | N.D. | Gayden et al.,[52] Kim et al.[72] |
| IFNAR1 | complete | recessive | ∼100% (lifetime) | N.D. | Bastard et al.[55] |
| IFNAR1 | hypomorphic | recessive | unknown | N.D. | Al Qureshah et al.[53] |
| IFNAR1 | hypomorphic | dominant | unknown | N.D. | Al Qureshah et al.[53] |
| IFNAR2 | complete | recessive | ∼100% (lifetime) | N.D. | Duncan et al.[56] |
| IKBKB | complete | recessive | 100% | N.D. | Pannicke et al.,[57] Cuvelier et al.[73] |
| MEFV | GoF | recessive | ∼100% (for p.Met694Val) | N.D. | French FMF Consortium,[59] The International FMF Consortium et al.,[60] Ben-Chetrit,[74] Gershoni-Baruch et al.[75] |
| MEFV | GoF | dominant | N.D. | RR = 6–8 | Cazeneuve et al.,[76] Medlej-Hashim et al.,[77] Jéru et al.,[78] Eyal et al.[79] |
| NOD2 | hypomorphic | recessive | 1.5% | OR = 10–42 | Ogura et al.,[21] Hugot et al.,[22] Hampe et al.,[23] Ahmad et al.,[64] Bonen et al.,[65] Yazdanyar et al.[80] |
| PTCRA (Pre-TCRα) | hypomorphic | recessive | ∼12% (Centogene cohort) unknown over lifetime | OR = 5 | Materna et al.[66] |
| TYK2 | hypomorphic | recessive | ∼80% (lifetime) | OR = 5 | Kerner et al.[68] |

HLA alleles are not included. N.D., no data.
[a]Calculated based on the incidence of SPTCL in Māori/Pacific people and the allele frequency in Polynesians.[52,72]

required for its formation. In cases of C9 deficiency, complement activity is severely reduced but not abolished. C9 deficiency is frequent in Japan,[93] where the p.Arg116* (rs121909592) loss-of-function variant has a frequency as high as 0.0298[46,94] (Table 1). In Japan, ∼1 person in 1,100 is predicted to have a complete deficiency of C9. Based on data for the city of Fukuoka in Japan, the annual risk of developing meningococcal disease is 1,000 times greater in C9-deficient individuals than in other individuals, with an annual penetrance of 0.1% in C9-deficient individuals versus 0.0001% in other individuals (Table 2).[43] The deleterious alleles of C6 and C9 common in African and Japanese populations may have provided an evolutionary advantage. The MAC is a double-edged sword; it contributes to antibacterial immunity and allows lysis of Gram-negative bacteria, but this lysis is accompanied by release of bacterial components, such as lipopolysaccharide, which can trigger inflammation and septic shock. Würzner hypothesized that partial C6 deficiency would have been an evolutionary advantage in the event of endotoxin shock during infection with Gram-negative bacteria.[95] This hypothesis could be extended to complete C6 deficiency in Africans, C9 deficiency in Japanese, and C2 deficiency in Europeans. Moreover, purging of these alleles

may be slow, as they are pathogenic only in the homozygous state. Nevertheless, it is intriguing that MAC protein deficiency confers a predisposition only to *Neisseria* meningitis, suggesting that the MAC is largely redundant against other pathogens, including other Gram-negative bacteria (e.g., *E. coli*, *B. pertussis*, and *V. cholerae*).

## Common variants of *MEFV* can underlie autoinflammation

Other family-based studies have focused on familial Mediterranean fever (FMF), the most frequent known genetic autoinflammatory disease worldwide, first described in 1908.[96] It was first shown to be a genetic disease in 1945,[97,98] and biallelic mutations of *MEFV* were reported in 1997.[59,60,99] FMF is caused by homozygosity or compound heterozygosity for gain-of function variants of *MEFV*, encoding pyrin,[59,60,74] with clinical penetrance being incomplete or complete depending on the genotype (Table 2).[75] Heterozygosity for *MEFV* variants may also be a significant risk factor for the development of FMF (RR ∼6–8 times higher than for non-carriers), as about a third of the patients carry only a monoallelic variant.[76–79] In heterozygous

patients, non-random autosomal monoallelic expression of the *MEFV* locus is warranted to be tested, as it might explain incomplete penetrance (Box 2). FMF may therefore be considered a semi-dominant condition with a lower risk and penetrance in heterozygotes. Pyrin promotes assembly of the pyrin inflammasome and interleukin-1 β (IL-1β) secretion.[100] Gain-of function *MEFV* variants impair the binding of the 14-3-3 inhibitory protein to pyrin, promoting uncontrolled inflammasome activation.[101,102] Excessive inflammation in the patients leads to recurrent fever accompanied by serositis (peritonitis, pleuritis, pericarditis, or synovitis). Left untreated, repeated flare-ups of inflammation can lead to secondary amyloidosis, which may cause serum amyloid A (SAA) protein deposition, resulting in kidney failure.[61]

FMF is particularly prevalent in people of Jewish, Armenian, Arabian, or Turkish descent, with the frequency of allele carriage estimated at 20% in these populations.[103–105] Four founding variants of exon 10 account for most cases in populations from the Mediterranean basin: p.Met680Ile (rs28940580), p.Met694Ile (rs28940578), p.Met694Val (rs61752717), and p.Val726Ala (rs28940579). For instance, the p.Val726Ala variant has an allele frequency of 0.039 in Ashkenazi Jews (gnomAD v.4.1) (Table 1). The p.Met694Val allele has a frequency of 0.017 in Turks (Turkish genome project) and 0.08 in North African Jews (Table 1).[62] The p.Met694Val allele is also associated with an increased risk of ankylosing spondylitis in Turks, with an OR of 4.8.[106] It has been suggested that the high prevalence of these alleles in Mediterranean and Jewish populations results from a selective advantage of heterozygosity, which is thought to have provided protection against severe infectious diseases of the past. It was recently suggested that heterozygosity for *MEFV* variants confers cellular resistance to *Yersinia pestis* by decreasing the interaction of MEFV with *Yersinia* outer protein M (YopM), a *Yersinia pestis* virulence factor, but preserving its binding to the WT human pyrin, thereby attenuating YopM-induced IL-1β suppression.[107–109] It is therefore tempting to speculate that heterozygosity for any of these *MEFV* variants may have conferred protection against epidemics of plague, which killed a very large proportion of people in Europe and the Middle East.

### Common variants of *ADA, DCLRE1C,* and *IKBKB* can underlie severe T cell deficiencies

Patient-based studies of a globally rare immunodeficiency paradoxically revealed causal alleles in small populations in which this deficiency is not that rare. T and B lymphocytes define adaptive immunity and are essential for long-term survival in an ever-changing environment containing a multitude of pathogens. T and B cells differentiate in the thymus and bone marrow, respectively, following a tightly regulated process involving the somatic and clonal rearrangement of the genomic loci corresponding to the T and B cell receptors.[110] Severe combined immunodeficiencies (SCIDs) are a group of Mendelian IEIs defined by a lack of T cell development, which may be associated with deficiencies of other lineages (e.g., natural killer [NK] and B cells).[111] In the absence of hematopoietic stem cell transplantation, SCIDs are invariably lethal due to overwhelming infections during the first year of life.

SCID is very rare in the general population (about 1 in 50,000), but 1 of 2,000 neonates of the Navajo and Apache Native American populations have T⁻B⁻NK⁺ SCID. This condition, known as Athabascan SCID (SCIDA) since 1980,[50] was estimated in 1991 to be driven by a single gene variant present in 2.1% of the corresponding population,[112] mapped to chromosome 10 in 1998.[71] The risk of SCIDA is increased by homozygosity for a single-nucleotide substitution in *DNA Cross-Link Repair 1C* (*DCLRE1C*) that was identified in 2002.[51] *DCLRE1C* is a crucial gene for T cell receptor (TCR) and B cell receptor (BCR) rearrangement during the differentiation of T and B cells, respectively.[113] The resulting p.Tyr199* variant (rs121908157) has an allele frequency of 0.021 in the Navajo and Apache populations (Tables 1 and 2). Similarly, T⁻B⁻NK⁻ SCID prevalence in Somali neonates is about 1 in 5,000 due to a common premature stop codon in the adenosine deaminase (ADA) gene.[39] ADA is an enzyme that catalyzes irreversible deamination of adenosine and deoxyadenosine to inosine and deoxyinosine and is required to prevent cellular toxicity, in particular in immature lymphocytes. The p.Gln3* variant (rs1057520217) has an allele frequency of 0.024 in Somali people and is hypomorphic. Homozygotes have partial ADA deficiency (Tables 1 and 2).[39,114] Of note, the allele frequency study was performed among Somali migrants in Denmark in 2007. Although an ascertainment bias is unlikely, a larger study in Somalia remains to be performed, as one cannot exclude the possibility that a specific Somali subpopulation emigrated to Denmark.

The nuclear factor κB (NF-κB) pathway plays a major role in signal transduction downstream of many receptors and is involved in many biological processes.[115] The *Inhibitor Of Nuclear Factor Kappa B Kinase Subunit Beta* (*IKBKB*) gene encodes the IKKβ molecule, which plays a major role in the NF-κB canonical pathway. Inherited IKKβ deficiency is associated with a SCID-like infectious phenotype but with normal T cell counts.[57,73] The patients have a combined immunodeficiency (CID). Interestingly, the first patients with IKKβ deficiency to be reported were all of Northern Cree descent and were living in remote communities in the Manitoba and Saskatchewan provinces of Canada.[57] All were homozygous for a frameshift mutation, p.Gln432Profs*62 (rs886041036), which was later shown to have an allele frequency of 0.076 in Northern Cree individuals[58,116] (Table 1). The Apache, Navajo, and Cree populations are relatively small, consisting of ~300,000, ~70,000, and ~200,000 people, respectively. The Somali population is larger, consisting of about 18 million people. DCLRE1C, ADA, and IKBKB deficiencies in these populations neatly illustrate the fact that alleles conferring a predisposition to severe infections can be found at relatively high frequency in specific, small, isolated human populations. The high frequency of these alleles in these populations is unlikely to result from balancing selection. It probably results from a founder effect; in other words, a genetic drift with isolation or bottlenecks followed by rapid expansion of the corresponding populations.[117] Its persistence attests to the slow purging of recessive conditions even when homozygotes die in infancy.

### A common *TYK2* variant underlies tuberculosis

Tuberculosis (TB) is an airborne disease typically triggered by *Mycobacterium tuberculosis*. TB is endemic in many countries

and the leading cause of death from a single pathogen.[118] Every year ~10 million people fall ill due to *M. tuberculosis* infection, and 1.3 million individuals die from TB (World health organization, 2022). Mendelian susceptibility to mycobacterial diseases (MSMD) was first described in 1951.[119] The patients are susceptible to Bacillus Calmette-Guérin (BCG) vaccine substrains and environmental mycobacteria. Defects of 22 genes, underlying 47 allelic forms, with autosomal-recessive (AR), X-linked recessive (XLR), and autosomal-dominant (AD) modes of inheritance have been described.[120–126] The causal genes are physiologically related, as almost all their products are involved in interferon-γ (IFN-γ)-mediated immunity. Rare IEIs identified as causal for MSMD have been found in patients without MSMD but with TB as their sole phenotype.[127] Two disorders in particular have been diagnosed in several patients with TB, AR complete IL-12Rβ1 and TYK2 deficiencies, both of which impair both IL-12- and IL-23-dependent IFN-γ immunity. As these two disorders are rare, with a frequency of less than $10^{-5}$ in the general population, they account for only a very small proportion of TB cases even though their penetrance for TB is higher than that for MSMD, *M. tuberculosis* being about 1,000-fold more virulent than BCG and environmental mycobacteria. Nevertheless, this has provided proof of principle that defects of IFN-γ-mediated immunity can underlie isolated TB in humans without MSMD.[128–130]

These initial findings led to the discovery that homozygosity for the common p.Pro1104Ala (rs34536443) variant of *TYK2* underlies TB in patients of European ancestry.[67] Homozygosity for p.Pro1104Ala also underlies MSMD but with a much lower penetrance than for TB. This variant impairs the IL-23 response pathway as profoundly as complete deficiency of TYK2, but selectively, resulting in impairment of IFN-γ production by specific lymphocyte subsets without any detectable impact on the IL-12 response pathway (or the IL-10 and type I IFN pathways).[130,131] The frequency of this variant in Europeans is 0.0445, leading to a prevalence of about ~1 in 600 for homozygosity (Table 1). This variant is absent in sub-Saharan Africa and very rare in East Asia and has a prevalence of about 1% in other regions. In a subsequent study focusing on a European population based on UK Biobank data,[132] homozygosity for p.Pro1104Ala was found to account for about 1% of cases of TB in British individuals, with an OR of developing TB of 5 in homozygous carriers relative to heterozygotes or non-carriers.[68] Lifetime penetrance for the development of TB upon infection was estimated at about 80% for homozygotes (Table 2). Remarkably, the p.Pro1104Ala allele was also shown to have a protective effect against two autoimmune diseases, rheumatoid arthritis and systemic lupus erythematosus, possibly also accounting for its high frequency.[133] The p.Pro1104Ala allele originates from a founder effect about 30,000 years ago in Western Eurasians.[134] This accounts for its high prevalence in Europe, its presence in populations with European admixture, its rarity in Eastern Asia, and its absence from sub-Saharan Africa. The frequency of this variant has decreased slowly but steadily in Europeans over the last 2,000 years, from 13% to 4%, implying that negative selection has occurred, consistent with the very high burden of TB in Europe.[134] About one billion Europeans are estimated to have died from TB in the last

2,000 years.[68,135,136] Thus, studies of MSMD led to the discovery of a common monogenic etiology accounting for about 1% of past and present cases of TB in humans of European descent.

### Common *IFNAR1* or *IFNAR2* variants can underlie viral diseases

Family-based studies of type I IFNs, a group of 16 IFNs (13 IFN-α genes, 2 of which encode identical proteins, IFN-β, IFN-ε, IFN-κ, and IFN-ω) binding a heterodimeric receptor composed of Interferon Alpha And Beta Receptor Subunit 1 (IFNAR1) and 2 (IFNAR2), have surprisingly led to discoveries of public health relevance.[137] Type I IFNs were first described in 1957 as molecules able to interfere with viral replication *in vitro*.[138] Upon binding to their receptor, type I IFNs induce a complex signaling cascade that plays a crucial role in antiviral immunity. IEIs impairing the type I IFN pathway underlie susceptibility to a narrow range of severe viral infections, including critical influenza pneumonia (e.g., IRF7, IRF9, and STAT2 deficiencies), critical COVID-19 pneumonia (e.g., IRF7, IFNAR1, IFNAR2, STAT2, and TYK2 deficiencies), herpes simplex virus 1 (HSV-I) encephalitis (e.g., IFNAR1), recurrent rhinovirus infection (e.g., IFIH1), and infections with live attenuated virus vaccines (e.g., IFNAR1, STAT1, and STAT2).[130,139–156] While the penetrance of individual viral infections is incomplete, most if not all patients suffer from at least one viral infection (Table 2). Autoantibodies against type I IFNs phenocopy IEIs of the IFN type I signaling pathway, further confirming the link between this pathway and sporadic severe viral infections.[152,157–159] Surprisingly, high frequencies of null alleles of *IFNAR1* and *IFNAR2* were recently reported in two geographically distant and isolated populations: Western Polynesians and Arctic people, respectively.[55,56,160] The p.Glu386* variant (rs201609461) was found with an allele frequency of 0.0125 in Western Polynesians, with a frequency of homozygosity estimated at ~1 in 6,500 in Samoans. The p.Ser53Pro variant (rs1987287426) of IFNAR2 is loss of function and was found with an allele frequency of 0.034 in Inuits from Greenland, Canada, and Alaska, with a homozygosity rate of ~1 in 1,500 Greenlanders. Homozygotes are prone to a small number of life-threatening viral illnesses. We cannot exclude the possibility that these variants provided a selective advantage to these populations in an unknown situation. However, in such geographically isolated populations, it appears more likely that this high allele frequency results from genetic drift with serial founder effects, isolation, or bottlenecks followed by rapid expansions of the population.[160,161]

Eleven hypomorphic IFNAR1 variants were recently identified.[53] These variants severely impair IFN-α and IFN-ω signaling but largely spare IFN-β signaling. Ten of these alleles are rare in all populations studied, but the remaining allele (p.Pro335del) is common in South Han Chinese, with an allele frequency ~0.02. As a result, it is predicted to be present in the homozygous state in ~1 of 2,500 individuals in this area[54] (Table 1). Homozygosity for the p.Pro335del allele was found to be associated with critical COVID-19 pneumonia in a 16-year-old patient. In addition, the hypomorphic variants show signs of negative dominance when co-expressed with the wild-type allele. Cells heterozygous for these variants display a dominant phenotype *in vitro*, with impaired responses to IFN-α and -ω but not -β, and viral

susceptibility. Consistent with these results, preliminary observations suggest that patients heterozygous for these variants are prone to respiratory and cerebral viral diseases with incomplete penetrance, attesting to both the dominance of these variants clinically and the importance of IFN-α and -ω for protective immunity against some respiratory and cerebral viruses. It is remarkable that a loss-of-function and dominant negative variant of IFNAR1 can reach such a high allele frequency. These results strongly suggest that the p.Pro335del variant is an important risk factor for severe viral infections in the South Han Chinese, at least when present in the homozygous state. With an estimated 16 million Chinese heterozygous for this variant, further population genetics studies are warranted to confirm the association of this variant with a higher risk of severe viral infections and estimate the clinical penetrance in both heterozygotes and homozygotes.

## PATHOGENIC ALLELES WITH HIGH RISK BUT LOW PENETRANCE

### Common variants at *HLA* loci underlie autoimmune conditions

Several human leukocyte antigen (HLA) alleles, some rare and some common, have been strongly associated with autoimmune conditions in population-based studies, conferring high risks with incomplete penetrance, with an OR >10 or even >50. These HLA-associated conditions may have been the evolutionary price to pay for the pathogen-driven positive selection of highly diverse HLA alleles over thousands of years.[162–164] As a trade-off, protection against infection early on, before or during reproductive age, may convey a risk of autoimmunity later in life, during or after the reproductive period. Alternatively, protection against infections in a given environment may convey a risk of autoimmunity in another environment, whether due to human migration to another region or environment modification in the same region. Both hypotheses can account for the dual phenotypes of individual humans but are more likely to manifest in populations over successive generations. Regardless, the larger the number of HLA alleles, the greater the diversity of microbial T cell epitopes presented to T cells in the infected population, and the higher the likelihood of at least some infected children and young adults surviving. In turn, the diversity of HLA alleles increases the risk that any given population may contain pathogenic variants at non-HLA loci controlling T cell tolerance to self and unleashing the recognition by T cells of self-peptides presented by HLA, the mechanism underlying clinical autoimmunity, particularly in middle-aged and elderly adults but not exclusively in these groups. This is the classic explanation of HLA-associated autoimmunity.

For example, HLA-B*57 (MAF ∼14%) and HLA-B*27 (MAF ∼8%) provide strong protection against disease progression in HIV-1-infected Europeans (OR = 7 and 3.4, respectively).[165,166] They may have protected children against other infections in the past. However, HLA-B*27 is also strongly associated with a higher risk of developing both ankylosing spondylitis (AS) (OR = 46) and post-infectious "reactive" arthritis (OR > 30).[167–170] Remarkably, AS penetrance is only 1.2% in HLA-B*27+ Europeans, whereas it is 21% in the HLA-B*27+ relatives of HLA-B*27+ AS patients, highlighting the considerable

impact of variants at other loci.[171] However, this risk is not restricted to adults, as two frequent haplotypes, DRB1*03-DQB1*0201 (DR3) or DRB1*04-DQB1*0302 (DR4), which have frequencies of 1%–30% in most human populations, have been known since the 1970s to be associated with a higher risk of developing a life-threatening condition of childhood type 1 diabetes (T1D) (OR = 3.6 and 8.4, respectively).[172–175] The highest risk is for DR3/DR4 individuals (genotype frequency in European controls ∼2% versus ∼30% in T1D; OR = 18).[176] The persistence of common HLA alleles underlying a life-threatening condition of childhood when present in the heterozygous state suggests that these alleles may have conferred a major protective advantage in the past, possibly in another environment; that their pathogenic impact is more recent; or that their frequency had been steadily declining until insulin therapy became available. Thus, both rare and common HLA alleles can underlie autoimmune conditions. The description of all of these conditions is beyond the scope of this review.[177–179]

### Common *NOD2* variants underlie Crohn's disease
Population-based studies have shown Crohn's disease — a chronic inflammatory bowel disease (IBD) characterized by patchy intestinal inflammatory lesions in the gastrointestinal tract leading to chronic abdominal pain, diarrhea, obstruction, and/or perianal lesions, can have a monogenic origin.[180] The prevalence of this disease is highest in North America, Western and Northern Europe, and Oceania but is increasing in other parts of the world, suggesting a strong impact of environmental factors.[180] Genome-wide linkage analyses have identified three frequent variants of *NOD2* (*nucleotide-binding oligomerization domain-containing 2*) collectively associated with an OR for disease development between 10 and 44 in homozygous or compound heterozygous carriers relative to controls.[21–23,64,65] The OR is much lower for heterozygotes (OR = 2.6). These three variants are p.Arg675Trp (rs2066844, also known as p.Arg702Trp), p.Gly881Arg (rs2066845, also known as p.Gly908Arg), and p.Leu980Profs*2 (rs2066847, also known as p.Leu1007Profs*2), with allelic frequencies ranging between 0.02 and 0.05 in Europeans and Middle Eastern individuals (Table 1). Despite the very strong association in multiple studies, a study of the Danish general population showed that the clinical penetrance of Crohn's disease at 50 years of age in biallelic carriers remains low, at ∼1.5% in homozygotes (Table 2).[23,80,181]

NOD2 may play a major role in regulation of the intestinal microbiota. It activates the NF-κB pathway in myeloid cells and ileal Paneth cells by recognizing the muramyl dipeptide (MDP) of intracellular bacterial lipopolysaccharides (LPSs).[182] NOD2 activation induces production of various cytokines, chemokines, and antimicrobial peptides in a cell type-dependent manner.[182] In particular, Paneth cells synthesize and secrete various antimicrobial peptides or proteins into the intestinal lumen, including lysozyme, human α-defensins 5 and 6 (HD5 and HD6, respectively), and secreted phospholipase A2 (sPLA2). In the mouse model, bacterial killing by Nod2-deficient ileal cells is impaired, resulting in ileal dysbiosis. This dysbiosis increases the interaction of bacterial products with immune cells, leading to chronic inflammation and the typical histological presentation, with transmural infiltration by lymphocytes

and macrophages, together with granuloma. The three frequent variants associated with Crohn's disease are hypomorphic, as they retain their ability to induce basal NF-κB activation in the absence of activation, but completely fail to activate NF-κB signaling in the presence of LPSs.[21,65] It has been suggested that their high allelic frequency in Europeans attests to a protective role against septic shock in carriers, as demonstrated in the mouse model.[183]

### A common CCR5 variant protects against HIV and confers a predisposition to West Nile virus infection

Population-based human genetic studies have investigated infection with human immunodeficiency virus 1 (HIV-1), a retrovirus primarily infecting CD4[+] T cells, leading to their progressive loss and, ultimately, to acquired immunodeficiency syndrome (AIDS).[184] CD4 is the receptor for HIV-1 on CD4[+] T cells, with C-C chemokine receptor 5 (CCR5) as the principal coreceptor. Autosomal-recessive CCR5 deficiency confers resistance to infection with HIV-1 with high, if not complete, penetrance.[47–49] Certain protective CCR5 alleles are rare (e.g., p.Cys101*, also known as c.303T>A or rs1800560),[185] but at least one loss-of-function allele of CCR5 is common: a 32-bp deletion leading to premature termination of protein synthesis (rs333, p.Ser185Ilefs*32, CCR5-Δ32). This allele has a MAF of ~0.1 in Europeans, among whom the rate of homozygosity is ~1% (Table 1). The CCR5-Δ32 allele is less frequent, or even absent, in African and Asian populations. Given the recent emergence of HIV-1, resistance to this virus cannot explain the high frequency of the CCR5-Δ32 allele in the European population. This allele emerged in Northern Europe at least 7,000 years ago, and stabilized to its modern frequency around 2,000 years ago, suggesting earlier selection events.[186] It has been suggested that there was intensive selection for the CCR5-Δ32 variant during ancient pandemics of diseases such as plague or smallpox.[187–189]

CCR5-deficient individuals are otherwise apparently normal, but two studies have suggested that they have a higher risk of symptomatic West Nile virus (WNV) disease (WNVD), with an OR of 5.9 in Europeans (Table 2).[24,25,70] The penetrance is unknown. Consistently, CCR5-deficient mice invariably develop fatal WNV encephalitis upon infection.[190] WNV is an RNA flavivirus transmitted by bites from infected mosquitoes. Clinical manifestations occur in only 20% of infected individuals and include fever, headache, tiredness, body aches, nausea, vomiting, skin rash, and swollen lymph glands. Less than 1% of infected individuals develop life-threatening WNV encephalitis. Remarkably, autoantibodies neutralizing type 1 IFNs underlie about 40% of cases of WNV encephalitis.[158] It may not, therefore, be coincidental that CCR5 is highly expressed on T cells and plasmacytoid dendritic cells (pDCs), the most potent type I IFN-producing cell types. CCR5 deficiency may impair optimal recruitment of pDCs to the site of infection. This hypothesis is plausible, as it would be consistent with the risk conferred by auto-antibodies against type I IFNs and with patients with various inherited or acquired conditions of T cells not prone to WNVD. As WNV has only recently reached the shores of Southern Europe, predisposition to lethal disease in homozygotes is consistent with the spread of the mutant CCR5 allele from Northern to Southern Europe due to elusive selective forces.

### A common variant of TIM3 underlies SPTCL with a high risk of HLH

Subcutaneous panniculitis-like T cell lymphoma (SPTCL), accounting for less than 1% of diagnosed non-Hodgkin's lymphomas, is characterized by the infiltration of CD8[+] αβ T cells into the subcutaneous adipose tissue, where they surround adipocytes in a lace-like pattern.[191–193] Affected individuals typically display multiple subcutaneous nodules, night sweats, fever, and weight loss, and, in ~20% of cases, associated autoimmune disorders, most commonly systemic lupus erythematosus.[194] In about 20% of cases, SPTCL is associated with hemophagocytic lymphohistiocytosis (HLH),[193] life-threatening hyperinflammation caused by uncontrolled activation of lymphocytes and macrophages. In their study of a series of 27 patients with SPTCL, Gayden et al. found that 60% of their patients carried germline biallelic loss-of-function variants of the hepatitis A virus cellular receptor 2 (HAVCR2) gene, which encodes T cell immunoglobulin mucin 3 (TIM3).[52] TIM3 is the third member of the TIM family. It is widely expressed across the leukocytes of the immune system. Its best characterized ligand is galectin-9, which is strongly expressed on myeloid cells and endothelial cells and in the gastrointestinal tract. Following binding, TIM3 acts as an immune checkpoint in the maintenance of self-tolerance and antitumoral immunity.[195,196] TIM-3-deficient SPTCL patients have an earlier age at onset of disease than other patients and an extremely high risk of developing HLH (30–80%).[52,197]

Intriguingly, most of the patients in the first series were from Polynesia and East Asia, and all carried the same variant, p.Tyr82Cys (rs184868814). Population genetics analysis revealed allele frequencies of 0.04 and 0.01656 (gnomAD v.3.1) in Polynesia and East Asia, respectively (Table 1).[52] These findings were replicated in other series.[197–199] Māori/Pacific individuals were shown to have a higher risk of SPTCL than Europeans (RR = 11),[72] whereas the OR in East Asian homozygotes was estimated at ~10,000 relative to East Asian heterozygotes or non-carriers.[198] With 1 in 625 Polynesians and 1 in 3,600 East Asians predicted to be homozygous for the loss-of-function p.Tyr82Cys variant, TIM3 deficiency clearly drives SPTCL with low penetrance (if all SPTCL cases in Polynesians are attributable to TIM3 deficiency, then the maximum penetrance in p.Tyr82Cys homozygotes, based on SPTCL incidence in Māori/Pacific individuals,[72] would be 0.34%/year and 20.4% over 60 years) (Table 2). However, the strength of the association is beyond reasonable doubt, and the breadth of the associated phenotypes remains to be assessed in large series of carriers. The very high frequency of the p.Tyr82Cys variant in Polynesians probably results from bottlenecks followed by expansion, but the reasons for its high frequency in East Asians remains unclear.

### Common PTCRA alleles can underlie autoimmunity

Patient-based studies of rare patients with pre-TCRα deficiency led to another observation of public health relevance. Adaptive immunity is defined by subsets of cells using rearranged antigen receptors, including αβ T cells, γδ T cells, and B cells. αβ and γδ T cells differentiate from bone marrow-derived progenitors in the thymus. Early thymocytes simultaneously rearrange their TCR δ, TCRγ and TCRβ loci. If early thymocytes successfully rearrange the TCRγ and TCRδ loci, then they differentiate into γδ T cells.

If they successfully rearrange the TCRβ, then the TCRβ chain is expressed at the cell surface in the pre-TCR complex thanks to its dimerization with the pre-TCRα constant chain (pTα; encoded by *PTCRA*). This process, known as β-selection, is essential for thymocyte survival and proliferation before TCRα rearrangement and differentiation into mature αβ T cells. Complete T cell differentiation defects are associated with SCID phenotypes (see common variants of *ADA*, *DCLRE1C*, and *IKBKB* can underlie severe T cell deficiencies), whereas partial T cell differentiation defects are associated with autoimmunity and a less severe susceptibility to infection. Mice lacking pTα have >95% fewer αβ T cells in the periphery. Ten humans homozygous or compound heterozygous for private or rare biallelic loss-of-function variants leading to complete pTα deficiency and impaired pre-TCR complex formation were recently reported.[66] These patients have a small thymus, profound αβ T cell lymphopenia in early life, and abnormally high counts of γδ T cells. Despite this severe biological phenotype, they remain asymptomatic until their teenage years or early adulthood and develop infections and autoimmunity. This late disease onset can be explained by the production of a small but sufficient number of functional αβ T cells providing protection against infectious diseases.

In the same study,[66] two hypomorphic *PTCRA* alleles present in the general population were also identified: p.Tyr76Cys (rs141630791) and p.Asp51Ala (rs200942121). Both alleles severely impair pre-TCR complex formation *in vitro*. The p.Tyr76Cys allele is found in people of African ancestry, with an allele frequency of 0.003517 (gnomAD v.4.1), and is predicted to be present in the homozygous state in 1 in 80,000 Africans, possibly more in specific African populations. The p.Asp51Ala variant is common in the Middle East and South Asia, with an allele frequency as high as 0.011 in Iran and 0.020 in Pakistan,[66] and is therefore predicted to be homozygous in 1 in 2,500–10,000 individuals in the corresponding populations (Table 1). Like mice with similar knockin mutations,[200] homozygous p.Asp51Ala carriers produce abnormally large numbers of γδ naive T cells, but their peripheral αβ T cell counts are normal.[66] In a large cohort, p.Asp51Ala homozygotes had a five times higher risk of developing autoimmunity than heterozygotes or non-carriers, with a disease penetrance of about 12% (Table 2). However, this risk and penetrance are probably underestimated because the cohort studied was young (mean age: 9.5 years), and the cumulative risk of developing autoimmunity increases with age. This variant may have provided a selective advantage in a specific, past environment. Alternatively, it may not be purged because it does not significantly decrease survival fitness before reproductive age, and even then only in homozygotes.

## DISCUSSION

Monogenic immunological disorders involving common variants have already been identified for 15 human loci (with *HLA* arbitrarily considered as a single locus). These genotypes were discovered in patient-based studies of rare (e.g., FMF) or common (e.g., TB) conditions (*MEFV*, *C2*, *C6*, *C9*, *DCLRE1C*, *ADA*, *IKBKB*, *TYK2*, *TIM3*, *IFNAR1*, *IFNAR2*, and *PTCRA*) and population-based studies of common conditions (*HLA*, *NOD2*, and

*CCR5*). After a pause between 2013 and 2018, their rate of discovery has recently accelerated (Figure 2). The first five common alleles underlying monogenic immunological disorders were identified in the last quarter of the 20th century (*HLA*, *MEFV*, *C2*, *C6*, and *C9*), another five were identified between 2000 and 2013 (*NOD2*, *DCLRE1C*, *ADA*, *CCR5*, and *IKBKB*), and the remaining five (*TYK2*, *TIM3*, *IFNAR1*, *IFNAR2*, and *PTCRA*) have been discovered since 2018. Remarkably, some of these genes carry several common variants that are pathogenic (*HLA*, *MEFV*, *NOD2*, and *IFNAR1*). Some of these alleles are common only in small, isolated populations, as best exemplified by a null *IKBKB* allele underlying SCID in homozygotes of Northern Cree decent.

By contrast, some of the other alleles are common in large populations, as illustrated by a severely hypomorphic *PTCRA* allele in the Middle East and South Asia, where homozygotes are prone to various types of autoimmunity. Most of these variants are recessive, causing disease only when present in the homozygous state. The frequency of at-risk genotype carriers (homozygous and compound heterozygous) is therefore much lower than the MAF. Admittedly, the MAF cutoff used to separate rare and common alleles (0.01) is arbitrary, with no significant genetic difference between a MAF of 0.009 and a MAF of 0.011. The definition of a population is also arbitrary, ranging from a hamlet to one of the seven major ancestries. Moreover, only a small proportion of populations of intermediate size and an even smaller proportion of smaller populations have been subject to sufficiently profound genetic analyses to estimate the most relevant MAF (i.e., the MAF of the allele in the smallest relevant population) with a reasonable degree of confidence. It therefore seems plausible that there are many more common alleles underlying monogenic immunological conditions.

Excluding *HLA*, which merits its own separate analysis, we identified 17 common variants at 13 loci. This number is relatively small next to the >450 monogenic IEIs due to rare variants discovered in patient-based studies. This probably results from the common practice of filtering out common alleles when searching for new IEIs – and monogenic inborn errors at large. We think that this notion has important implications for future research in biology and medicine. Some population-based studies originally detected common variants of single genes as disease causing (*HLA*, *NOD2*, and *CCR5*), but only a few have tested the hypothesis of recessive inheritance. This aspect should probably be reconsidered, particularly for very large studies with high statistical power. This notion also has important implications for patient-based studies. These findings suggest that, during genetic study of a patient or group of patients, the population to which they belong should be defined as accurately as possible. For example, three unrelated patients from France, Germany, and Italy would have Western Europe as a common denominator. The populations of the corresponding countries should also serve individually as reference populations. Use of the province of origin (e.g., Brittany, Piedmont, and Bavaria) would provide even higher granularity. The MAF of candidate alleles and the prevalence of the phenotype of interest should be defined at these different levels. The genetic data for each patient should ideally be considered in the context of their smallest, relevant population. The MAF of an allele in the "human

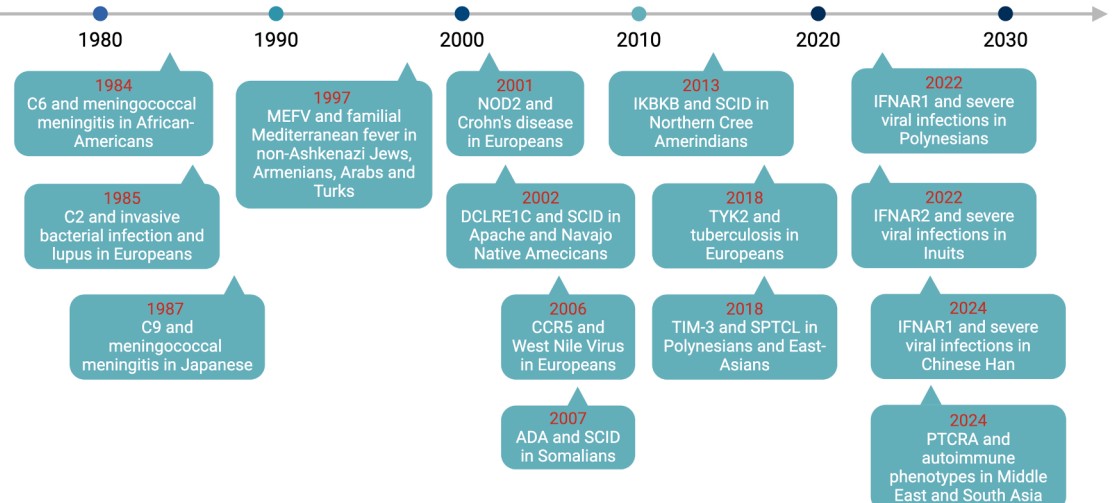

**Figure 2. Timeline of the discovery of common IEIs to infections**
We indicate the time point when the common molecular defect was associated with the indicated phenotype. Some of the molecular or immunological defects were known for decades before the association between the genotype and the phenotype was established.

ADA: ADA deficiency was identified in two sporadic SCID cases in 1972,[201] the first mutations in ADA were reported in 1986,[202] the first Somalian patient with the p.Gln3* variant was identified in 1995,[114] and the high prevalence of partial ADA deficiency in Somalians was recognized in 2007.[39]

C2: C2 deficiency was identified in sporadic cases in 1966,[33] the high prevalence of complete C2 deficiency in Europeans was reported in 1985,[34] and the corresponding variant was discovered in 1992.[35]

C6: C6 deficiency was identified in sporadic cases in 1974,[87] the high prevalence of complete C6 deficiency in Afro-Americans was reported in 1984,[41] and the corresponding variant was discovered in 1996.[42]

C9: C9 deficiency was identified in sporadic cases in 1979,[92] the high prevalence of complete C9 deficiency in the Japanese was reported in 1987[93] and its association with meningococcal diseases in 1989,[43] and the corresponding variant was identified in 1998.[94]

CCR5: CCR5 deficiency and its relationship to resistance to HIV-1 infection were discovered simultaneously in 1996.[47–49] The higher risk of developing symptomatic WNV infection in CCR5-deficient individuals was discovered in 2006.[24]

DCLRE1C (ARTEMIS): The high prevalence of SCID in Apache and Navajo populations was first described in 1980.[50] The prevalence of heterozygous carriers in these populations was estimated at 2.1% in 1991.[112] The causal mutation of *DCLRE1C* was identified in 2002.[51]

HAVCR2 (TIM-3): complete TIM-3 deficiency and its association with SPTCL and HLH were described together in 2018.[52]

IFNAR1: complete IFNAR1 deficiency and its association with severe viral infections were discovered in 2019.[142] The discovery of a frequent loss-of-function variant in Polynesians was reported in 2022.[55] A frequent hypomorphic variant in South Chinese Han was reported in 2024.[53]

IFNAR2: complete IFNAR2 deficiency and its association with severe viral infections were discovered in 2015.[140] A frequent loss-of-function variant in Inuits was reported in 2022.[56]

IKBKB: A high prevalence of IKBKB deficiency was detected in the Northern Cree in 2013, corresponding to the first description of human IKBKB deficiency.[57] This observation was later confirmed by newborn screening.[58]

MEFV: familial Mediterranean fever (FMF) was first described in 1908.[96] The high prevalence of FMF in Jewish individuals and Armenians was reported in 1945.[97] The corresponding frequent variants of the *MEFV* gene were identified 1997.[59,60]

NOD2: NOD2 deficiency and its association with Crohn's disease were described together in 2001.[21–23]

PTCRA: the hypomorphic PTCRA variant frequent in South Asia and the Middle East was identified together with the rare complete loss-of-function variants.[66] The frequent hypomorphic variant is associated with autoimmune phenotypes with incomplete penetrance.

TYK2: complete TYK2 deficiency was first described in 2006[203] and its association with viral and mycobacterial diseases in 2015.[129] A common variant underlying partial TYK2 deficiency and susceptibility to TB in Europeans was discovered in 2018.[67]

general population" is not informative enough for rigorous analysis of a patient's exome or genome.

In this light, we think that common alleles should not be systematically filtered out when considering phenotypes that have a local or global prevalence between $10^{-4}$ and $10^{-2}$ for recessive traits and even perhaps $10^{-5}$. This emphasizes the importance of ascertaining population prevalence of a given immunological disorder, which should match the MAF of fully penetrant pathogenic alleles (e.g., SCID in Navajo and Apache populations). Nevertheless, incomplete penetrance should be considered (Table 2), which may account for an apparent mismatch between disease prevalence and MAF of pathogenic alleles. In total, 14 common alleles are currently known to be autosomal and to un-

derlie recessive phenotypes, and another three are known to be semi-dominant (two variants of *MEFV* and one of *IFNAR1*) (Table 1). However, AD and XLR modes of inheritance should also be considered. For XLR and AD disorders, phenotypes with a higher prevalence, up to $10^{-2}$, can be considered.

The evolutionary basis of the surprising commonality of the variants described in this review is mostly hypothetical and probably varies from case to case. For instance, population bottlenecks followed by rapid expansion are the most likely hypothesis explaining the high frequency of SCID-causing *IKBKB* or *DCLRE1C* variants in Native American populations. A similar mechanism probably also explains the high frequency of null *IFNAR1* and *IFNAR2* alleles in Polynesians and Arctic people,

respectively. The reasons for the high frequency of a severely hypomorphic and dominant-negative *IFNAR1* allele in Southern Chinese are less clear. Recessive alleles are counterselected only in individuals carrying two deleterious copies and may therefore persist at high frequencies for longer periods of time, particularly if they display incomplete penetrance or are associated with a mild clinical phenotype. This is probably the case for the p.Pro1104Ala *TYK2* allele, whose the frequency has declined in Europeans from 13% to 4%, over the last 3,000 years due to the selective pressure exerted by TB on homozygotes.

A third mechanism is balancing selection due to beneficial or pathogenic properties of the variant depending on zygosity. The best example is perhaps provided by variants of *MEFV* for which heterozygosity is thought to protect against plague, whereas homozygosity underlies FMF. Likewise, C2, C6, and C9 deficiencies confer predisposition to invasive bacterial infections, whereas the variants concerned may protect against septic shock in heterozygotes. In addition, the same genotype may have opposite effects, depending on the microbial environment, which itself varies over time and space. For example, homozygosity for variants causing CCR5 deficiency protects against HIV-1 infection and may have protected against other unknown pathogens in the past, but it also confers a predisposition to WNVD. Likewise, variants of *NOD2*, *PTCRA*, *TYK2*, or *TIM3* may have become common due to genetic drift before the advent of recent environmental triggers that revealed their pathogenicity. For instance, the recent rapid, global increase in the prevalence of Crohn's disease in homozygotes for pathogenic *NOD2* variants may be due to the recent spread of ad hoc environmental cues.

These findings also have practical implications. When a new monogenic disorder due to rare alleles is discovered, systematic experimental studies of the other alleles at the same locus should be performed. This is important not only to ensure that the cumulative frequency of the deleterious genotype, whether loss of function or gain of function, is consistent with the prevalence of the phenotype studied but also because it may lead to the discovery that the same genotype and phenotype are less rare, and perhaps even common, in another small or even large population. Moreover, this approach may reveal that common variants have different biochemical impacts, raising the possibility that they underlie another phenotype that may (e.g., if common alleles are hypomorphic and the rare alleles are loss of function) or may not (e.g., if common alleles are gain of function [GoF] or neomorphic and the rare alleles are loss-of-function) be related.

The *PTCRA* gene neatly illustrates this point, as the discovery of ultra-rare alleles underlying complete deficiency of pre-TCRα in homozygotes with overt clinical immunodeficiency led to the discovery of an almost complete deficiency in hundreds of thousands of South Asians and Middle Eastern individuals with various isolated forms of autoimmunity and only a mild immunological phenotype. Before the advent of whole-exome and whole-genome sequencing, the search for IEIs only rarely took into account the MAF of candidate alleles, let alone the cumulative MAFs of the deleterious alleles of a gene. Sequencing of 50 or 100 healthy controls, rarely of homogenous or relevant ancestry, was often deemed sufficient. Nowadays, thanks to large public databases (e.g., gnomAD),[36] it is easy to obtain access to information about rare and common alleles in all major ancestries and even increasingly in more specific populations. Experimental testing of all alleles has become a requirement, with discovery of common and deleterious alleles revealing the breadth of consequences of monogenic lesions in multiple populations.

Finally, these findings imply that monogenic lesions should be considered in patients with clinical phenotypes that are not as rare as those typically associated with IEIs and that such lesions can be discovered not only in population-based studies but also in patient-based studies that are extended into population-based studies. A significant proportion of cases of "common diseases" may be caused by monogenic IEIs and common alleles. This is neatly illustrated by the *PTCRA* alleles underlying autoimmunity and the *TYK2* allele underlying TB, both of which highlight the importance of considering common alleles for recessive traits, not just, as traditionally, for dominant or semi-dominant (additive) traits.

The genetic architecture of common diseases may thus be revisited by considering monogenic common lesions. Recessive traits should be considered in genome-wide association studies (GWASs) of common variants; they may be highly penetrant in a subset of or the entire population sample. Candidate monogenic genotypes may then be investigated experimentally through targeted functional and familial genetic studies. Moreover, statistical significance does not equal biological significance; the TYK2 p.Pro1104Ala variant is not statistically significant when tested by GWAS for association with TB in a recessive model because of the correction for multiple testing. However, it is clearly causal when computational and experimental lines of evidence are considered together. The limitations imposed by the need to correct for multiple testing and by the experimental difficulties to test non-coding common variants are intrinsic to the GWAS approach. Common immunological diseases may be due to various recessive or dominant traits involving common alleles in a significant proportion of patients. Based on these findings and this model, it is likely that many more patients worldwide than previously thought suffer from monogenic disorders. The findings described in this review suggest that there may be millions of people with monogenic immunological disorders. They suggest that forward genetic approaches may not be sufficient to appreciate fully the impact of single-gene lesions. Reverse genetic approaches are probably warranted for experimental testing of all alleles at specific loci before computational and agnostic assessments of the clinical impact of biochemically deleterious alleles in different populations.

### ACKNOWLEDGMENTS

We would like to thank all members of the HGID laboratory for fruitful discussions, particularly Laurent Abel, Aurélie Cobat, and Jérémie Rosain for critical reading of an earlier version of this paper. This work was supported in part by the St. Giles Foundation; the Rockefeller University; Institut National de la Santé et de la Recherche Médicale (INSERM); the Imagine Institute; Paris Cité University; the National Center for Research Resources; the National Center for Advancing Sciences of the National Institutes of Health (NIH) (UL1TR001866); the NIH (R01AI088364, R01AI095983, R01AI163029, and U19AI162568); the American Lung Association (COVID-1026207); the Stavros

Niarchos Foundation (SNF) as part of its grant to the SNF Institute for Global Infectious Disease Research at The Rockefeller University; the Square Foundation, Grandir – Fonds de solidarité pour l'enfance; the Fondation du Souffle; the SCOR Corporate Foundation for Science; the Battersea and Bowery Advisory Group; the French National Research Agency (ANR) under the "Investments for the Future" program (ANR-10-IAHU-01); the Integrative Biology of Emerging Infectious Diseases Laboratory of Excellence (ANR-10-LABX-62-IBEID); ANR GENVIR (ANR-20-CE93-003); ANR AAILC (ANR-21-LIBA-0002); ANR AI2D (ANR-22-CE15-0046); ANR MAFMACRO (ANR-22-CE92-0008); ANR GENFLU (ANR-22-CE92-0004); ANR PTCRA (ANR-24-CE15-5334); the ANR-RHU COVIFERON program (ANR-21-RHUS-08); the French research agency on infectious and emerging diseases (ANRS) project ECTZ170784-ANRS0073; the Horizon-HLTH-2021-DISEASE-04 program under grant agreement 101057100 (UNDINE); the European Union's Horizon 2020 Research and Innovation Program under grant agreement 824110 (EASI-genomics); the French Foundation for Medical Research (Equation 201903007798); the French foundation for cancer research (ARC) project AR-CAGEING2022040004944 and ARCPGA2024110008994_9650, Robert Debré Association for Medical Research, W.E. Ford, General Atlantic's Chairman and Chief Executive Officer, G. Caillaux, General Atlantic's Co-President, Managing Director, and Head of Business in EMEA, and the General Atlantic Foundation; the French Ministry of Higher Education, Research, and Innovation (MESRI-COVID-19); and REACTing-INSERM.

## DECLARATION OF INTERESTS

The authors declare no competing interests.

## SUPPLEMENTAL INFORMATION

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
