## [Document S1. Transparent peer review records for Béziat et al. · Cell Genomics]

Cell Genomics, Volume 6

Supplemental information

Monogenic disorders of immunity:

Common variants are not so rare

Vivien Béziat and Jean-Laurent Casanova

Monogenic disorders of immunity: common variants are not so rare

Vivien Béziat and Jean-Laurent Casanova

Summary

Initial submission: Received : Jan 14, 2025

Scientific editor: Sara Rohban

First round of review: Number of reviewers: 3
Revision invited : Apr 16, 2025
Revision received : Aug 25, 2025

Second round of review: Number of reviewers: 2
Revision invited : Oct 07, 2025
Revision received : Oct 17, 2025

Third round of review: Number of reviewers: 1
Accepted : Dec 10, 2025

Data freely available: NA

Code freely available: NA

This transparent peer review record is not systematically proofread, type-set, or edited. Special characters, formatting, and equations may fail to render properly. Standard procedural text within the editor's letters has been deleted for the sake of brevity, but all official correspondence specific to the manuscript has been preserved.

Referees' reports, first round of review

Reviewer #1:

The authors set out to challenge the current paradigm that monogenic inborn errors of immunity (IEI) can only result from rare (<1%) pathogenic alleles. This MAF is indeed a commonly applied threshold when interrogating sequencing data for IEI-relevant pathogenic alleles in either clinical diagnostic or research contexts. The subject matter is therefore important and the review timely, given recent discoveries. It is also provocative and arguably misleading in certain areas, so I would urge the authors to consider the following points if revising their manuscript.

1. The majority of the review is given over to the discussion, one by one, of "common" alleles putatively linked to monogenic IEI, in chronological order of their discovery. In my view, a more helpful structure would instead group and organise these variants to highlight shared reasons underlying the apparently paradoxical association of common alleles with IEI that have been assumed to be rare. The abstract teases the reader with a numbered list of potential mechanisms - however these themes are not systematically developed within the review that follows.
2. The authors adopt an inclusive approach that embraces incompletely penetrant risk alleles for relatively prevalent conditions such as Crohn's disease (NOD2) or "autoimmunity" (PTCRA) alongside 100% penetrant pathogenic alleles for extreme and rare immunological phenotypes including SCID/CID (DCLRE1C, IKBKB). At the same time, GWAS is unreasonably dismissed, as for example in the final paragraph: "The limitation imposed by the need to correct for multiple testing and the lack of experimental support are intrinsic and inevitable limitations of the GWAS approach". I am missing a statistical genomic analysis that places these genetic risks of varying effect size within a unifying framework that respects the complementarity of alternative investigative methods.
3. Related to the above, there is almost no discussion of the importance of accurately ascertaining population prevalence of a given immunological disorder in relation to the maximum MAF expected for a fully penetrant pathogenic allele. That pathogenic alleles of MEFV or Artemis deficiency are common in those populations that also suffer much higher prevalence of the corresponding diseases is entirely expected.
4. When it comes to incomplete penetrance, the choice of a lower limit OR of 5 for a "monogenic" effect seems very low and just as arbitrary as the upper MAF threshold of 1% with which the authors take issue. The whole concept of penetrance is problematic when it comes to susceptibility to infectious disease, dependent as it must be on exposure to the relevant pathogen. Does it matter what proportion of sufferers from a particular "disease" bear pathogenic alleles of a designated gene and/or whether variants in that gene alone are sufficient to confer disease, independently of other risk factors? I think there is a broader discussion to be had here about what constitutes a monogenic disease and the utility of identifying genetic drivers, both for the individual/family and in terms of scientific understanding. Badging incompletely penetrant HLA-associated autoimmune diseases as "monogenic" does not seem helpful.

Reviewer #2:

Drs. Beziat and Casanova review the topic of Common genetic variants associated with clinically relevant functional effects. They compare these with the conventional concept that rare genetic variants are more likely to be the cause of clinical disease. They focus the review immune diseases, their area of expertise, but the concept is relevant to any group of genetic diseases where the pathogenicity of rare vs. common genetic variants is an area of ongoing controversy and study.

Overall, the review is thorough, well referenced, and generally well written. There are a couple of rough sections that don't flow as well as they could and revisions of the text are suggested below. I appreciate that for most of the disorders the review that are associated with common variants, they provide both a historical perspective of when the disease was described and when the gene association was made complete with citations of original papers. For most sections, the authors also speculate about potential events/conditions that could have driven selection for or against each common variant.

I'm well acquainted with the concept of penetrance but the description of penetrance in the 2nd and 3rd sentences of the introduction is perplexing and required multiple reads to understand what the authors were trying to convey. It's a rough way to start an otherwise excellent review and I'm concerned that readers may be discouraged from continuing on into the remainder of the review. I'd suggest that the authors revisit and try to clarify/simplify this section.

The section of the introduction on page 3 that gives 6 potential reasons for incomplete penetrance is an important conceptual section of the review and sets up the background for the detailed descriptions of each disorder that follows. This would lend itself to a figure that succinctly summarizes these 6 reasons/concepts. Please consider adding - I think it would enhance the paper.

In the 2nd paragraph of section 1 "Common variants at HLA loci underlie autoimmune conditions" the authors discuss the risks of AS in HLA-B27 positive individuals and suggest it may have been selected for due to a potential benefit in responses to infections. Since the question of infection responses is raised here, I'd suggest that the authors add a comment that in addition to the increased risk for AS, HLA-B27 positivity dramatically increases the risk of post-infectious reactive arthritis.

In section 3 "C6 and C9 deficiency", the 2nd paragraph begins "C9 deficiency of the paragraph that begins "C9 Deficiency was first described...." the second sentence reads "Indeed, during the formation of the CAM..." I think this was intended to be "the MAC" not the CAM - double-check

In the last paragraph of the paper's discussion section, the sentence "Auto-Abs against cytokines were already known to affect millions of people" seems out of place here and may be either deleted or moved."

Reviewer #3:

In the manuscript, Beziat and Casanova present a review of known monogenic disorders of immunity that are caused by common alleles. An important concept that is usually not considered when these group of disorders are studies. The paper is well researched, comprehensive, and clearly written. The message is important and timely. However, the introduction lacks clarity and the discussion can be expanded to help the different groups of researchers involved in the filed collaborate more closely across methodological and genetic architecture boundaries. Additionally, a number of references to previous work are missing. Please find my recommendations below.

Introduction: I find the introduction factually correct but unfocused and hard to follow. The authors do not get to the main point of the paper (high MAF variants that can cause IEIs) until page four! Even though they bring lots of examples that (if read by a - fully awake- genetics expert) can lead to their main points. I think the introduction can be revised and tightened. Some suggestions are:

- Even though I agree with the authors arguments around penetrance [Gaining an understanding of ... SMAD6 alleles] I think this section is loosely connected with the rest of the introduction. Is the idea to imply incomplete penetrance is common in monogenetic diseases? Or that is is one possible cause that pathogenic alleles can rise to high MAF? If the first, I think this section can be added in a box in the paper because it is not directly related to the question of MAF. If the later this should be stated clearly and also added to the abstract and the rest of the paper and one of the reasons underlying high-MAF pathogenic variants.
- Similarly, the section on the RR and OR, while correct it diverts from the main focus of the paper what is the point of this section and how does it relate to the main focus of the paper? What is the main conclusion form that section as it relates to the main point of the review? Is the example necessary in the text or the RR, OR example can be put into box within the paper?
- All in all, the message of pages 2 and 3 seems to be not all pathogenic variants are fully penetrant or have high effect sizes. If I try to add personal knowledge to this and read in between the lines, this rationally leads to: so not all pathogenic variants are under heavy negative selection nor all of them are pathogenic in all contexts and thus they can reach high MAFs. If this is the message the authors can state it sooner and clearer to avoid a diluted rational or hidden links that the reader must excavate.
- The fact that neither of these two points (penetrance and effect size) is part of the four reasons mentioned in the abstract adds to the confusion. Should they be added? How do these points relate to bottlenecks, drift, slow purging, and balancing selection when it comes to high MAF pathogenic variants in IEI?

Individual examples:

- HLA: References 25-27 do not include more recent finding on the relationship between HLA and possible selection conferred by pathogens in ancient humans. Please add these references individually, one example include <https://doi.org/10.1038/s41586-023-06618-z>
- C2: Needs references: "Patient-based studies led to the discovery of common genetic defects of complement", "Genetic deficiencies of MAC components underlie invasive disease due to Neisseria.", "Complete deficiencies of alternative pathway proteins confer a predisposition to invasive bacterial infections"

- NOD2: "Population-based studies have shown Crohn's disease — a chronic inflammatory bowel disease (IBD) characterized by patchy intestinal inflammatory lesions in the gastrointestinal tract leading to chronic abdominal pain, diarrhea, obstruction and/or
 - perianal lesions — to have a monogenic origin". Monogenic CD cases are a small % of total patient population. The sentence is misleading as is, please adjust to make clear most CD cases are in fact polygenic/complex. The paper the authors reference makes this clear and explicitly says in the abstract "Crohn's disease is a complex disease".
 - SCID: "The high frequency of these alleles in Native American populations is unlikely to result from balancing selection. It probably results from genetic drift, with isolation or bottlenecks followed by rapid expansion of the corresponding populations." This is known as founder effect, please add to this section. (see: <https://www.pnas.org/doi/full/10.1073/pnas.0903341106>).
 - Type I interferon: The examples of IEI impairing the type I IFN pathway lack references susceptibility to common respiratory viruses found before the ones in relation to covid-19. Please add the relevant references (<https://pubmed.ncbi.nlm.nih.gov/28716935/> and <https://pubmed.ncbi.nlm.nih.gov/28606988/>).
- Missing example: G6PD deficiency and its relationship with diabetes in African and African American populations is another example that fits with the subject of this paper. Please add. (For more information see: <https://pubmed.ncbi.nlm.nih.gov/38918629/>).

Conclusion: I agree with the authors that non-additive models, particularly recessive models, should be considered in GWAS. However, I believe GWAS deserves a more expanded discussion. While GWAS has inherent limitations, some of which the authors mention, it remains a powerful tool for identifying causal variants, genes, and loci, when combined with appropriate functional follow up - particularly for the types of variants discussed in this paper. This is increasingly true in light of the rapid growth in GWAS sample sizes and diversity, driven by expanding biobank resources and decreasing data generation costs. Furthermore, modern GWAS no longer rely solely on minor allele frequency thresholds but instead use minor allele count cutoffs, which allow for well-calibrated test statistics and improved power, aligning with the authors' argument for avoiding arbitrary MAF cutoffs.

There are numerous examples of GWAS supporting genes or variants implicated in inborn errors of immunity (IEI), independent of IEI-focused studies. For instance, many HLA alleles have been identified through GWAS; TYK2 variants linked to autoimmunity were discovered before and outside of its known role in TB; and a recent study associated MVEF with ankylosing spondylitis in Middle Eastern populations (<https://pubmed.ncbi.nlm.nih.gov/30946743/>). I encourage the authors to explore their list of 14 genes/loci in the GWAS Catalog, they will find additional supporting examples.

Finally, some of the challenges related to conducting ancestry- or population-specific analyses, highlighted by the authors, have already been addressed within the GWAS and admixture mapping fields. Established approaches for accounting for genetic similarity and diversity in multi-ancestry cohorts can be adapted to patient-based studies as well.

All in all, expanding the discussion to include the complementarity of GWAS, patient/family-based studies, and experimental approaches would provide a more accurate representation of the field's evolution. It would also underscore how discovery rates can accelerate when researchers collaborate across methodological and genetic architecture boundaries. The authors are well positioned to highlight this important perspective.

Additional comments:

- Please number your pages and lines, it is really difficult to comment on a long manuscript without appropriate styling to help the reviewers.
 - In the introduction the authors state "...although there is no need to explain healthy cases with the detrimental genotype if all sick relatives carry an at-risk genotype that can be mechanistically connected with immunological and clinical phenotypes." While I agree than healthy individuals can carry putative or established causal variants, I disagree with this view that there is no need to further investigate the underlying causes of the variability in phenotype. Trying to understand this phenomenon has extended our understanding of genome structure and function for example the effect of regulatory variants in limiting shaping the penetrance of pathogenic coding variants (see 10.1038/s41588-018-0192-y). Please revise this sentence.
 - "A disease can be considered monogenic, even with low penetrance, provided that it is strongly associated with a monogenic genotype, with this association supported by experimental evidence." Please add appropriate references, one possible supporting study can be 10.1001/jama.2021.23686 which shows that most pathogenetic variants have indeed small ORs/effect sizes.
 - "However, a handful of monogenic immunological conditions were discovered in large population-based studies. In these studies, focusing on common conditions, the involvement of a common allele was expected." Please add references.
-

Authors' response to the first round of review

Authors: We warmly thank the reviewers for their positive and constructive evaluation. We revised our manuscript taking their helpful and insightful comments into account.

Reviewer #1: The authors set out to challenge the current paradigm that monogenic inborn errors of immunity (IEI) can only result from rare (<1%) pathogenic alleles. This MAF is indeed a commonly applied threshold when interrogating sequencing data for IEI-relevant pathogenic alleles in either clinical diagnostic or research contexts. The subject matter is therefore important and the review timely, given recent discoveries.

Authors: We warmly thank the reviewer.

It is also provocative and arguably misleading in certain areas, so I would urge the authors to consider the following points if revising their manuscript.

1. The majority of the review is given over to the discussion, one by one, of "common" alleles putatively linked to monogenic IEI, in chronological order of their discovery. In my view, a more helpful structure would instead group and organise these variants to highlight shared reasons underlying the apparently paradoxical association of common alleles with IEI that have been assumed to be rare. The abstract teases the reader with a numbered list of potential mechanisms - however these themes are not systematically developed within the review that follows.

Authors: We thank Reviewer #1 for this thoughtful suggestion. We agree that grouping genotypes based on the evolutionary causes of their elevated frequencies—if these causes were known with certainty—would be valuable for readers. We had initially considered such a classification in an earlier version of our review. For example, the high frequency of at-risk alleles in IKBKG, DCLRE1C, ADA, IFNAR1, and IFNAR2 probably reflects historical bottlenecks followed by rapid population expansion (founder effects). In contrast, the elevated frequency of at-risk MEFV and CCR5 variants is probably due to past balancing selection, providing a selective advantage to heterozygous carriers against an as-yet uncertain pathogenic threat. However, for other at-risk genotypes (C2, C6, C9, NOD2, PTCRA, TYK2, TIM3), the evolutionary causes remain unclear; we thus prefer not to assign them to specific categories. Instead, we revised our paper by grouping them in two main categories based on disease penetrance in at-risk genotype carriers (see below). In addition, in the introduction, we now describe the possible mechanisms explaining the high frequency of the deleterious alleles.

2. The authors adopt an inclusive approach that embraces incompletely penetrant risk alleles for relatively prevalent conditions such as Crohn's disease (NOD2) or "autoimmunity" (PTCRA) alongside 100% penetrant pathogenic alleles for extreme and rare immunological phenotypes including SCID/CID (DCLRE1C, IKBKB). At the same time, GWAS is unreasonably dismissed, as for example in the final paragraph: "The limitation imposed by the need to correct for multiple testing and the lack of experimental support are intrinsic and inevitable limitations of the GWAS approach". I am missing a statistical genomic analysis that places these genetic risks of varying effect size within a unifying framework that respects the complementarity of alternative investigative methods.

Authors: We thank reviewer #1 for raising this important point. As stated in the introduction, there is no universally accepted "monogenic threshold". While it is true that the variants in NOD2, PTCRA, TIM3, and CCR5 included in our review are far from being completely penetrant for the related immunological phenotypes, they confer odd ratios (OR) above our arbitrary threshold (>5), and causality is further supported by experimental evidence. As pointed by reviewer #1 our threshold is inclusive: we do not restrict our review to Mendelian disorders (i.e. monogenic with full penetrance). To our knowledge, beyond NOD2 and HLA alleles, no alleles identified in GWAS studies or populationbased studies of immune-related phenotypes reach our OR threshold of 5. Would the reviewers know relevant examples. we would be glad to include them. In addition, we toned down the sentence quoted by the reviewer, and reorganized our manuscript, with variants displaying high penetrance and risk vs. variants displaying low penetrance but high risk.

3. Related to the above, there is almost no discussion of the importance of accurately ascertaining population prevalence of a given immunological disorder in relation to the maximum MAF expected for a fully penetrant pathogenic allele. That pathogenic alleles of MEFV or Artemis deficiency are common in those populations that also suffer much higher prevalence of the corresponding diseases is entirely expected.

Authors: Thank you for this important suggestion. We now stress this important notion in the revised discussion of our manuscript.

4. When it comes to incomplete penetrance, the choice of a lower limit OR of 5 for a "monogenic" effect seems very low and just as arbitrary as the upper MAF threshold of 1% with which the authors take issue. The whole concept of penetrance is problematic when it comes to susceptibility to infectious disease, dependent as it must be on exposure to the relevant pathogen. Does it matter what proportion of sufferers from a particular "disease" bear pathogenic alleles of a designated gene and/or whether variants in that gene alone are sufficient to confer disease, independently of other risk factors? I think there is a broader discussion to be had here about what constitutes a monogenic disease and the utility of identifying genetic drivers, both for the individual/family and in terms of scientific understanding. Badging incompletely penetrant HLA-associated autoimmune diseases as "monogenic" does not seem helpful.

Authors: The definition of a monogenic effect is indeed important and challenging. As discussed in our revised introduction, we define monogenic inheritance based on a strong association (OR >5), which encompasses HLA-associated autoimmune diseases. This is admittedly inclusive. It does not exclude epistasis as a possible hypothesis for explaining low penetrance of some risk alleles. The possible mechanisms— in other words the "other risk factors"—are now discussed in Box 2, together with the mechanisms explaining incomplete penetrance.

Reviewer #2: Drs. Beziat and Casanova review the topic of Common genetic variants associated with clinically relevant functional effects. They compare these with the conventional concept that rare genetic variants are more likely to be the cause of clinical disease. They focus the review immune diseases, their area of expertise, but the concept is relevant to any group of genetic diseases where the pathogenicity of rare vs. common genetic variants is an area of ongoing controversy and study.

Overall, the review is thorough, well referenced, and generally well written. There are a couple of rough sections that don't flow as well as they could and revisions of the text are suggested below. I appreciate that for most of the disorders the review that are associated with common variants, they provide both a historical perspective of when the disease was described and when the gene association was made complete with citations of original papers. For most sections, the authors also speculate about potential events/conditions that could have driven selection for or against each common variant.

Authors: We wamly thanks the referee.

1. I'm well acquainted with the concept of penetrance but the description of penetrance in the 2nd and 3rd sentences of the introduction is perplexing and required multiple reads to understand what the authors were trying to convey. It's a rough way to start an otherwise excellent review and I'm concerned that readers may be discouraged from continuing on into the remainder of the review. I'd suggest that the authors revisit and try to clarify/simplify this section.

Authors: We thank reviewer #2 for this suggestion. Indeed, the previous version of our introduction was too complex. We have revised our introduction extensively, including the first sentences on penetrance. See below for more details.

2. The section of the introduction on page 3 that gives 6 potential reasons for incomplete penetrance is an important conceptual section of the review and sets up the background for the detailed descriptions of each disorder that follows. This would lend itself to a figure that succinctly summarizes these 6 reasons/concepts. Please consider adding - I think it would enhance the paper.

Authors: Thank you for this suggestion. Creating a figure summarizing these 6 different concepts is difficult, as each of them would deserve an independent figure. Instead we created a box summarizing these concepts (Box 2), as suggested by reviewer #3. This modification is in line with the first comment of reviewer #2. We think it improves our introduction.

3. In the 2nd paragraph of section 1 "Common variants at HLA loci underlie autoimmune conditions" the authors discuss the risks of AS in HLA-B27 positive individuals and suggest it may have been selected for due to a potential benefit in responses to infections. Since the question of infection responses is raised here, I'd suggest that the authors add a comment that in addition to the increased risk for AS, HLA-B27 positivity dramatically increases the risk of post-infectious reactive arthritis.

Authors: We thank reviewer #2 for this great suggestion. We now include a comment on the increased risk of reactive arthritis in HLA-B27 positive carriers.

4. In section 3 "C6 and C9 deficiency", the 2nd paragraph begins "C9 deficiency of the paragraph that begins "C9 Deficiency was first described...." the second sentence reads "Indeed, during the formation of the CAM..." I think this was intended to be "the MAC" not the CAM - double-check.

Authors: We thank the reviewer for pointing this typo. It is now corrected.

5. In the last paragraph of the paper's discussion section, the sentence "Auto-Abs against cytokines were already known to affect millions of people" seems out of place here and may be either deleted or moved." **Authors:** We agree with reviewer #2, the sentence was deleted.

Reviewer #3: In the manuscript, Beziat and Casanova present a review of known monogenic disorders of immunity that are caused by common alleles. An important concept that is usually not considered when these group of disorders are studied. The paper is well researched, comprehensive, and clearly written. The message is important and timely.

Authors: We warmly thank the reviewer.

However, the introduction lacks clarity and the discussion can be expanded to help the different groups of researchers involved in the field collaborate more closely across methodological and genetic architecture boundaries. Additionally, a number of references to previous work are missing. Please find my recommendations below.

1. Introduction: I find the introduction factually correct but unfocused and hard to follow. The authors do not get to the main point of the paper (high MAF variants that can cause IEs) until

page four! Even though they bring lots of examples that (if read by a - fully awake- genetics expert) can lead to their main points. I think the introduction can be revised and tightened.

Authors: We thank the referee for this very helpful comment, which is in agreement with comment 1 of reviewer #2. We extensively revised and trimmed the introduction, while adding two Boxes.

Some suggestions are:

2. Even though I agree with the authors arguments around penetrance [Gaining an understanding of ... SMAD6 alleles]] I think this section is loosely connected with the rest of the introduction. Is the idea to imply incomplete penetrance is common in monogenetic diseases? Or that is is one possible cause that pathogenic alleles can rise to high MAF? If the first, I think this section can be added in a box in the paper because it is not directly related to the question of MAF. If the later this should be stated clearly and also added to the abstract and the rest of the paper and one of the reasons underlying high-MAF pathogenic variants.

Authors: We thank reviewer #3 for this great suggestion. We now include the possible causes of incomplete penetrance in Box 2.

3. Similarly, the section on the RR and OR, while correct it diverts from the main focus of the paper what is the point of this section and how does it relate to the main focus of the paper? What is the main conclusion form that section as it relates to the main point of the review? Is the example necessary in the text or the RR, OR example can be put into box within the paper?

Authors: We thank reviewer #3 for yet another great suggestion. We now include the section on size effect, OR and RR, as well as the relationship between RR and penetrance, in Box 1.

4. All in all, the message of pages 2 and 3 seems to be not all pathogenic variants are fully penetrant or have high effect sizes. If I try to add personal knowledge to this and read in between the lines, this rationally leads to: so not all pathogenic variants are under heavy negative selection nor all of them are pathogenic in all contexts and thus they can reach high MAFs. If this is the message the authors can state it sooner and clearer to avoid a diluted rational or hidden links that the reader must excavate.

Authors: We thank reviewer #3 for this suggestion. We now clearly state in the revised introduction why evolutionarily the alleles of interest can be at high MAFs in a given population.

5. The fact that neither of these two points (penetrance and effect size) is part of the four reasons mentioned in the abstract adds to the confusion. Should they be added? How do these points relate to bottlenecks, drift, slow purging, and balancing selection when it comes to high MAF pathogenic variants in IEI?

Authors: We thank reviewer #3 for this suggestion. We now include the penetrance in the sentence relative to the slow purging.

Individual examples:

6. HLA: References 25-27 do not include more recent finding on the relationship between HLA and possible selection conferred by pathogens in ancient humans. Please add these references individually, one example include <https://doi.org/10.1038/s41586-023-06618-z>

Authors: We thank the reviewer for pointing us to this interesting research topic. We now quote a few example of positive selection at the HLA locus, including the suggested reference.

7. C2: Needs references: "Patient-based studies led to the discovery of common genetic defects of complement", "Genetic deficiencies of MAC components underlie invasive disease due to Neisseria.", "Complete deficiencies of alternative pathway proteins confer a predisposition to invasive bacterial infections".

Authors: We now include references for these three statements.

8. NOD2: "Population-based studies have shown Crohn's disease — a chronic inflammatory bowel disease (IBD) characterized by patchy intestinal inflammatory lesions in the gastrointestinal tract leading to chronic abdominal pain, diarrhea, obstruction and/or perianal lesions — to have a monogenic origin". Monogenic CD cases are a small % of total patient population. The sentence is misleading as is, please adjust to make clear most CD cases are in fact polygenic/complex. The paper the authors reference makes this clear and explicitly says in the abstract "Crohn's disease is a complex disease".

Authors: It is true that highly penetrant variants underlying inflammatory bowel diseases are rare, and were identified by patient based studies (e.g. IL10RB). Other variants, discovered by GWAS, have weak penetrance, sometime with an high OR, like NOD2. To our knowledge, the reason underlying the weak penetrance of NOD2 remains unknown. Genetic heterogeneity does not rule out the possible monogenic origin of a disease. It could be due to epistasis (digenic or polygenic), but experimental evidence is lacking. Disease penetrance in these patients may only require environmental factors, in line with an increasing disease prevalence worldwide.

Nevertheless, to clarify this point, we decided to split our review in two main sections:

- the pathogenic alleles with high penetrance
- the pathogenic alleles with high risk but low penetrance.

In addition, we tone down the quoted sentence to "Population-based studies have shown Crohn's disease — a chronic inflammatory bowel disease (IBD) characterized by patchy intestinal inflammatory lesions in the gastrointestinal tract leading to chronic abdominal pain, diarrhea, obstruction and/or perianal lesions — can have a monogenic origin."

This being said, we respectfully disagree with the referee's statement that "most CD cases are in fact polygenic/complex". These terms come from the field of population genetics and do not easily apply to clinical genetics. Whether any patient suffers from polygenic CD remains hypothetical: it has never been proven. As per complex CD, the lack of unambiguous definition of 'complex' prevents it to be tested.

9. SCID: "The high frequency of these alleles in Native American populations is unlikely to result from balancing selection. It probably results from genetic drift, with isolation or bottlenecks followed by rapid expansion of the corresponding populations." This is known as founder effect, please add to this section. (see: <https://www.pnas.org/doi/full/10.1073/pnas.0903341106>).

Authors: We thank the referee for this very helpful addition. We have modified the sentence accordingly as "The high frequency of these alleles in these populations is unlikely to result from balancing selection. It probably results from a founder effect, in other words a genetic drift, with isolation or bottlenecks followed by rapid expansion of the corresponding populations".

10. Type I interferon: The examples of IEI impairing the type I IFN pathway lack references susceptibility to common respiratory viruses found before the ones in relation to covid-19. Please add the relevant references (<https://pubmed.ncbi.nlm.nih.gov/28716935/> and <https://pubmed.ncbi.nlm.nih.gov/28606988/>).

Authors: We thank the referee for pointing this omission. We include these two references in our revised manuscript.

11. Missing example: G6PD deficiency and its relationship with diabetes in African and African American populations is another example that fits with the subject of this paper. Please add. (For more information see: <https://pubmed.ncbi.nlm.nih.gov/38918629/>).

Authors: We again thank the reviewer. This is an interesting example of yet another frequent disease-causing variant.

Variants underlying Glucose-6-phosphate dehydrogenase (G6PD) deficiency are divided in 5 main classes: Class 1--severe enzyme deficiency associated with chronic nonspherocytic hemolytic anemia.

Class 2--severe enzyme deficiency (less than 10%) associated with acute hemolytic anemia.

Class 3--moderate to mild enzyme deficiency (10-60%).

Class 4--very mild or no enzyme deficiency (60%).

Class 5—increased enzyme activity.

Only G6PD class 1 deficiency is considered as an IEI, and is included in the IUIS classification—within the congenital defects of phagocyte number or function.

The frequent variant in African and African American populations— known as A⁻ (Val68Met + Asn126Asp)—belongs to Class 3 G6PD deficiency. This is the reason why we did not integrate G6PD variants in our review.

12. Conclusion: I agree with the authors that non-additive models, particularly recessive models, should be considered in GWAS. However, I believe GWAS deserves a more expanded discussion. While GWAS has inherent limitations, some of which the authors mention, it remains a powerful tool for identifying causal variants, genes, and loci, when combined with appropriate functional follow up - particularly for the types of variants discussed in this paper. This is increasingly true in light of the rapid growth in GWAS sample sizes and diversity, driven by

expanding biobank resources and decreasing data generation costs. Furthermore, modern GWAS no longer rely solely on minor allele frequency thresholds but instead use minor allele count cutoffs, which allow for well-calibrated test statistics and improved power, aligning with the authors' argument for avoiding arbitrary MAF cutoffs.

Authors: We understand the referee's point. While we doubt that population-based approaches can do better than family-based approaches to discover human genotypes underlying severe phenotypes, we do not know the future and we understand and respect the referee's hypotheses and hopes. Nevertheless, due to space constraints, and because the review is focused on monogenic disorders, we opted to not expand the discussion of GWAS. Yet, we revised our text to sound more agnostic.

13. There are numerous examples of GWAS supporting genes or variants implicated in inborn errors of immunity (IEI), independent of IEI-focused studies. For instance, many HLA alleles have been identified through GWAS; TYK2 variants linked to autoimmunity were discovered before and outside of its known role in TB; and a recent study associated MVEF with ankylosing spondylitis in Middle Eastern populations (<https://pubmed.ncbi.nlm.nih.gov/30946743/>). I encourage the authors to explore their list of 14 genes/loci in the GWAS Catalog, they will find additional supporting examples.

Authors: We thank the referee for this very helpful suggestion. We mined the GWAS catalog for 15 loci, as we added ADA deficiency in Somalia in our revised manuscript. A few genotype/phenotype associations had an OR above 5. However, these connections were not with immune phenotypes, and were therefore not included in our review (ADA and aseptic loosening; HAVCR2 and Oligodendroglioma; IFNAR1 and glioblastoma; IFNAR2 and Familial squamous cell lung carcinoma; IFNAR2 and Type 2 diabetes). As pointed by reviewer #3, we also found an association between the rs61752717 in MEFV and ankylosing spondylitis. Although this association has an OR of 4.8, we now quote it in our review.

14. Finally, some of the challenges related to conducting ancestry- or population-specific analyses, highlighted by the authors, have already been addressed within the GWAS and admixture mapping fields. Established approaches for accounting for genetic similarity and diversity in multi-ancestry cohorts can be adapted to patient-based studies as well.

Authors: We thank the referee for raising this point. We agree that adjustment to ancestry is helpful for various types of genetic association studies.

15. All in all, expanding the discussion to include the complementarity of GWAS, patient/family-based studies, and experimental approaches would provide a more accurate representation of the field's evolution. It would also underscore how discovery rates can accelerate when researchers collaborate across methodological and genetic architecture boundaries. The authors are well positioned to highlight this important perspective.

Authors: We thank the referee for this important point. We think that the referee's point is perfectly valid, yet goes probably beyond the scope of this review, which is focused on common variants underlying monogenic IEI. In the future, we hope to write another review, ideally with a

population geneticist, which might help bridge the gap between patient- and population-based approaches.

Additional comments:

16. Please number your pages and lines, it is really difficult to comment on a long manuscript without appropriate styling to help the reviewers.

Authors: We thank the referee for this suggestion. We now include the page and line numbers.

17. In the introduction the authors state "...although there is no need to explain healthy cases with the detrimental genotype if all sick relatives carry an at-risk genotype that can be mechanistically connected with immunological and clinical phenotypes." While I agree that healthy individuals can carry putative or established causal variants, I disagree with this view that there is no need to further investigate the underlying causes of the variability in phenotype. Trying to understand this phenomenon has extended our understanding of genome structure and function for example the effect of regulatory variants in limiting shaping the penetrance of pathogenic coding variants (see 10.1038/s41588-018-0192-y). Please revise this sentence.

Authors: We apologize for the misunderstanding, probably accounting for the complexity of our previous introduction, as pointed by all three reviewers. We agree with the reviewer regarding the importance of investigating the healthy carriers of at risk genotypes. We wrote in the previous and current version of the paper: "Gaining an understanding of the mechanisms underlying the incomplete penetrance of monogenic disorders is a major endeavor in the field of IEI." We also want to thank the reviewer for pointing us to this interesting study. Although this 2018 paper adds weight to the 1997 hypothesis (McGee et al, AJHG; doi: 10.1086/301614), The proposed mechanism is not directly supported by evidence within a well-characterized disease context.. We nevertheless cite these two papers among the proposed causes of incomplete penetrance.

18. "A disease can be considered monogenic, even with low penetrance, provided that it is strongly associated with a monogenic genotype, with this association supported by experimental evidence." Please add appropriate references, one possible supporting study can be 10.1001/jama.2021.23686 which shows that most pathogenetic variants have indeed small ORs/effect sizes.

Authors: We thank the reviewer for pointing us to this interesting manuscript. We are not convinced by the authors' claims, as their study encompasses a very broad range of both loci and alleles. While some genes and alleles have been mechanistically and causally characterized in great depth, most genes and alleles have not. It is therefore difficult to claim that most pathogenic variants have low penetrance. It is more accurate to say that most alleles claimed to be pathogenic, with insufficient evidence, do not display high penetrance and even that some or perhaps many might not be pathogenic at all. We nevertheless cite this paper in the corresponding section.

19. "However, a handful of monogenic immunological conditions were discovered in large population-based studies. In these studies, focusing on common conditions, the involvement of a common allele was expected." Please add references.

Authors: We thank the referee and now include references.

Referees' reports, second round of review

Reviewer #3:

Thank you for addressing the concerns raised. One area I would urge you to revise more strongly is the treatment of GWAS. While your caution about its limitations is valid, the current tone can come across as dismissive. Given the central role GWAS plays in human genetics, a more balanced framing — one that acknowledges its complementarity with family-based and mechanistic studies, would make the manuscript both more accurate and more broadly useful to readers across disciplines.

Authors' response to the second round of review

Authors: We thank again the reviewers for their positive and constructive evaluation during the revision process.

Reviewer #3: Thank you for addressing the concerns raised. One area I would urge you to revise more strongly is the treatment of GWAS. While your caution about its limitations is valid, the current tone can come across as dismissive. Given the central role GWAS plays in human genetics, a more balanced framing — one that acknowledges its complementarity with family-based and mechanistic studies, would make the manuscript both more accurate and more broadly useful to readers across disciplines.

Authors: We do not mean do be dismissive about GWAS studies. After carefully reading the manuscript we have modified two sentences.

1. Page 4, line 63:

“Most monogenic disorders of immunity were discovered via patient- and family-based studies or studies of rare conditions in the field of IEI. **However, a handful of monogenic immunological conditions were discovered in large population-based studies.**” This sentence was replaced by:

“Most monogenic disorders of immunity were discovered via patient- and family-based studies or studies of rare conditions in the field of IEI. **However, a subset of monogenic immunological conditions were discovered in large population-based studies**”

2. Page 27, line 576:

“Excluding HLA, which merits its own separate analysis, we have identified 17 common variants at 13 loci. This number is relatively small next to the >450 monogenic IEI due to rare variants discovered in patient-based studies. **However, it is a relatively large number next to the number of monogenic conditions discovered in population-based studies, as all such attempts led to the discovery of a common allele.**” This sentence was replaced by: “Excluding HLA, which merits its own separate analysis, we have identified 17 common variants at 13 loci. This number is relatively small next to the >450 monogenic IEI due to rare variants discovered in patient-based studies. **This probably results from the common practice of filtering out common alleles when searching for new IEI – and monogenic inborn errors at large.**”

Authors' response to the third round of review

We have now added an additional sentence in the discussion outlining how population-based GWAS approaches and family-based studies can be integrated to identify and validate novel genotype–phenotype associations:

“Recessive traits should be considered in genome-wide association studies of common variants (GWAS); they may be highly penetrant in a subset of, or the entire population sample. Candidate monogenic genotypes may then be investigated experimentally, through targeted functional and familial genetic studies.”